



# Estimation of the Paris NO$_x$ Emissions from mobile MAX-DOAS observations and CHIMERE model simulations using the closed integral method

Reza Shaiganfar[1], Steffen Beirle[1], Hugo Denier van der Gon[2], Sander Jonkers[2], Jeroen Kuenen[2], Herve Petetin[3], Qijie Zhang[3], Matthias Beekmann[3], and Thomas Wagner[1]

[1]Max-Planck-Institute for Chemistry, Mainz, Germany

[2]TNO, dept. Climate, Air and Sustainability, Utrecht, The Netherlands

[3]LISA/IPSL, UMR CNRS 7583, Université Paris Est Créteil et Université Paris Diderot, Créteil, France

*Correspondence to*: Thomas Wagner (thomas.wagner@mpic.de)

**Abstract.** We determine NO$_x$ emissions for Paris in summer 2009 and winter 2009/2010 by applying the closed integral method (CIM) to a large set of car Multi-AXis (MAX)-DOAS measurements performed in the frame of the MEGAPOLI project. MAX-DOAS measurements of the tropospheric NO$_2$ vertical column density (VCD) are performed at large circles around Paris. From the combination of the observed NO$_2$ VCDs with wind fields the influx into and the outflux from the encircled area is determined. The difference of both fluxes represents the total emission. Compared to previous applications of the CIM, the large number of measurements during the MEGAPOLI campaign allowed the investigation of important aspects of the CIM. In particular the applicability of the CIM under various atmospheric conditions could be tested. Another important advantage of the measurements during MEGAPOLI is that simultaneous atmospheric model simulations with high spatial resolution (3 x 3 km²) are available for all days. Based on these model data it is possible to test the consistency of the CIM and to derive information about favorable or non-favorable conditions for the application of the CIM. We find that in most situations the uncertainties and the variability of the wind data dominate the total error budget. Also measurement gaps and uncertainties in the partitioning ratio between NO and NO$_2$ are important error sources. Based on a consistency check, we deduced a set of criteria on whether measurement conditions are suitable or not for the application of the CIM. We also developed a method for the calculation of the total error budget of the derived NO$_x$ emissions. Typical errors are between ±30% and ±50% for individual days (with one full circle around Paris). From the application of the CIM to car MAX-DOAS observations we derive daily average NO$_x$ emissions for Paris of 4.2 · 10$^{25}$ molecules/s for summer and of 7.8 · 10$^{25}$ molecules/s in winter. These values are by a factor of about 1.4 and 2.0 larger than the corresponding emissions derived from the application of the CIM to the model data, using the TNO-MEGAPOLI emission inventory, in summer and winter, respectively. Similar ratios (1.5 and 2.3 for summer and winter, respectively) were found for the comparison with the MACC-III emission inventory.

## 1 Introduction

Emission estimates of atmospheric trace species are important as input for model simulations and for the quantification of air pollution. Such emissions can be quantified using bottom-up or top-down techniques. Here we apply a 'local' top-down approach, the closed integral method (CIM), based on car MAX-DOAS measurements in combination with wind information. For the quantification of emissions, car MAX-DOAS measurements are performed on large circles around large cities or other strong emissions sources (Rivera et al., 2009; Ibrahim et al., 2010; Wagner et al., 2010; Shaiganfar et al., 2011, 2015). In contrast to top-down approaches based on satellite observations (e.g. Ghude eta al., 2013 and references therein),





emission estimates based on car MAX-DOAS measurements are independent from model simulations. They also depend much less on assumptions about the atmospheric lifetimes. Moreover, car-MAX-DOAS measurements are much less affected by clouds and aerosols than satellite observations.

Our study focusses on car MAX-DOAS observations during two extended measurement campaigns at Paris (Shaiganfar et al., 2015) in the framework of the European project MEGAPOLI (Baklanov et al., 2010; Beekmann et al., 2015, see also http://megapoli.dmi.dk/). During the first campaign in summer 2009 $NO_x$ emissions could be quantified on 9 days. During the second campaign in winter 2009/2010 $NO_x$ emissions could be quantified on 22 days. Another important aspect of this study is that highly resolved (3 x 3 km²) model simulations were available for all days of the car MAX-DOAS measurements. Thus, compared to previous studies, which are based on only a few days of car MAX-DOAS observations, the comprehensive set of car MAX-DOAS observations during both MEGAPOLI campaigns are well suited to address several important questions:

A) What are the uncertainties of $NO_x$ emission estimates based on the application of the CIM to car MAX-DOAS observations?

B) Which measurement settings (e.g. driving route) and conditions (e.g. wind speed and direction) are favorable, and which should be avoided?

C) How can the method further be improved?

D) How representative are the derived emissions for a specific time of the day or for the daily average?

E) What are the total NOx emissions from Paris during either summer or winter? How consistent are they with existing emission inventories?

The paper is organised as follows: In section 2 both MEGAPOLI campaigns and the car MAX-DOAS measurements are introduced. Section 3 gives an overview on the CHIMERE model. In section 4 the different steps of our approach are described in detail and the associated errors are discussed and quantified. Section 5 describes a consistency check of the method based on model simulations, and gives an overview on the derived $NO_x$ emissions for Paris during both campaigns. A summary and conclusions are provided in section 7.

## 2 MEGAPOLI campaigns and car MAX-DOAS measurements

The car MAX-DOAS observations in and around Paris were described in detail in Shaiganfar et al., 2015. Here we give only a brief overview. Two extensive measurement campaigns were organised in the frame of the MEGAPOLI project (Baklanov et al., 2010; see also http://megapoli.dmi.dk/) in June and July 2009 and in January and February 2010. Car MAX-DOAS measurements were performed on 25 days in summer and 29 days in winter. One major aim of the car MAX-DOAS measurements was to quantify the total $NO_x$ emissions from Paris. For that purpose we applied the closed integral method (CIM) by carrying out car MAX-DOAS measurements of the tropospheric vertical column density (VCD) along closed circles around Paris. Details on the data analysis of the car MAX-DOAS measurements are given in Shaiganfar et al. (2015), who used the same data set for comparison with satellite and model data.

On some days, the driving routes were not well suited for the determination of $NO_x$ emissions, because the driving routes did not cover full circles, and/or the circles were too small (they covered only the city center). On 9 days in summer 2009 and 22 days in winter 2009/2010 meaningful emission estimates were possible. In Fig. 1 a measurement example for 12 February 2010 is shown. This example represents almost ideal conditions, because the measurements were performed around a rather large circle without major gaps. Also, as indicated by the arrows, the surface-near wind speed (for details see section 4.1) was rather large (about 8.5 m/s) and the wind speed and wind direction did not change strongly during the





period of the measurements. The transport of polluted air masses towards the south-west is indicated by the enhanced $NO_2$ VCDs observed at the lower left part of the circle.

**3 CHIMERE model simulations**

The CHIMERE CTM (Schmidt et al., 2001; Menut et al., 2013) (www.lmd.polytechnique.fr/chimere) has been developed

since 1997 by IPSL (Institute Pierre Simon Laplace) and INERIS (Institut National de l'Environnement Industriel et des Risques). Simulations are performed with a horizontal resolution of 3 x 3 km² and a vertical discretization comprising eight vertical layers from ground to about 5 km, with decreasing vertical resolution with altitude. The TNO-MEGAPOLI inventory (Denier van der Gon et al., 2011; Timmermans et al., 2013) for the Paris region is a combination of a regional European emission inventory and a local emission inventory for Paris made by AirParif (2010), the city's air quality and emission

inventory authority. The base year of both inventories is 2005. In the TNO-MEGAPOLI inventory the year 2005 emissions for Paris (Ile de France) have been replaced with those from AirParif for the same year. The $NO_x$ emissions by month by source sector are shown in Fig 2 (top). Traffic is the largest $NO_x$ source in Paris. $NO_x$ emissions in winter are about 20% larger than in Summer due to the seasonal cycle of residential, commercial and other combustion processes.

An example of the spatial distribution and the diurnal variation of the $NO_x$ emissions over Paris is shown in Fig. 2 bottom. It

should be noted that since the replacement of Paris emissions with the Ile-de-France inventory from AirParif causes a change in the total emissions for France, the national total emissions would no longer be consistent with the official reported emissions. Therefore the difference in emissions has been attributed to the whole country except for the Ile-de-France region (per pollutant and emission source category). In Figure A1 in the appendix the nesting of a the high resolution inventory in the coarser scale regional inventory is shown. For the Paris region the emission data are available at 1km resolution, but for

the CHIMERE simulations they were averaged to the spatial resolution of the model (3km). Meteorological data are produced at hourly time steps with the PSU/NCAR Mesoscale Model (MM5; Dudhia et al., 1993) (see Fig. 3). More information about the specific CHIMERE simulations during MEGAPOLI can be found in Zhang et al. (2013 and 2015), Petetin et al. (2015) and Shaiganfar et al. (2015).

**4 Determination of the $NO_x$ emissions**

**4.1 Calculation of the $NO_2$ fluxes**

The $NO_x$ flux from the encircled area is calculated in several steps. In the first step the $NO_2$ fluxes are integrated using the closed integral method (CIM).

$$F_{NO_2} = \oint_S VCD(s) \cdot \vec{\omega} \cdot \vec{n} \cdot ds \qquad (1)$$

Here VCD indicates the tropospheric vertical column density of $NO_2$ (the vertically integrated $NO_2$ concentration), $\vec{\omega}$ indicates the wind vector and $\vec{n}$ the unit vector orthogonal to the driving route. Since individual MAX-DOAS measurements are performed for limited periods of time (about 1 min) the integral is substituted by a sum over the fluxes calculated for the segments corresponding to individual measurements:





$$F_{NO_2} \approx \sum_i VCD_i \cdot \omega_i \cdot \sin(\beta_i) \cdot \Delta s_i \qquad (2)$$

Here $VCD_i$ represents the $NO_2$ VCD obtained from measurement i, $\omega_i$ represents the average wind speed during the period of the car MAX-DOAS measurements. $\beta_i$ represent the angle between the (average) wind direction and the driving direction at the location of measurement I, $\Delta s_i$ indicates the distance between the location of measurement i and i+1.

As wind data we use the wind fields from the model simulations at different heights (see section 3). From this data set we calculate the average wind speed and direction over the measurement area during the whole time period of the car MAX-DOAS measurements along the circle. Average wind data are chosen (instead of using spatio-temporally resolved wind data), because usually 3D high-resolution wind fields are not available for the locations of the car-MAX-DOAS measurements. The effect of the use of spatio-temporally resolved wind data (interpolated in time and space to the individual car MAX-DOAS

observations) is usually small (except for days with high wind variability. It is investigated in detail in section 4.6.3.

As an example wind data for 24 January 2010 is shown in Fig. 3. On this day rather large variations of the wind speed and direction were found during the period of the car MAX-DOAS measurements: the wind speed at the surface varied by about a factor of two; the wind direction at the surface changed by about 20°. For most days smaller variations of both quantities are found.

For the calculation of the $NO_2$ fluxes we average the wind data from the PSU/NCAR Mesoscale Model (MM5; Dudhia et al., 1993)  weighted by an exponentially decreasing profile with scale heights of 300m (winter) and 500m (summer). These altitude ranges are rough estimates for trace gas profiles close to emission sources and take into account the effect of different vertical mixing conditions and atmospheric liftetimes in different seasons.  On individual days, however, also substantial deviations from these assumptions can be found (see Fig. A1 in the appendix). Nevertheless, changes of the wind fields with

altitude are typically smaller at higher altitudes. Thus the effect of the assumed profile height has a rather small influence on the derived wind fields. The effect of uncertainties of the wind speed and direction are discussed in section 4.6.3. We also compared the wind fields from the MM5 model with wind profiles measured by a cube lidar at the SIRTA site at Palaiseu in the South-West of Paris (Haeffelin et al., 2005) The comparison was possible for three altitude layers between 40 m and 200 m. We compared the averaged wind speed and wind directions during the periods of the MAX-DOAS measurements (see

Fig. A 2 in the appendix). For the layers at 120 m and 200 m, almost perfect agreement is found between both data sets. However, for the surface layer the model data systematically underestimate the wind speed obtained by the LIDAR. We have no clear explanation for these differences, but probably they are related to the limited horizontal resolution of the model data. Fortunately, the differences of the wind speed become much smaller (typically < 0.3 m/s) when averaged over the $NO_x$ layer heights of 300 m and 500 m, respectively.

**4.2 Effect of measurement gaps along the circles**

Measurement gaps can occur due to various reasons. Besides missing spectra due to instrumental problems also the quality of some measured spectra might be not good enough for a meaningful data analysis (e.g. due to over- or undersaturation caused by obstacles like trees, bridges or tunnels). In Fig. 4 measurements for 23 July 2009 are shown. On this day several gaps are present. The effect of gaps on the emission estimates can be particularly large if strong gradients of the trace gas

concentrations are present. This is e.g. the case for some of the gaps shown in Fig. 4 as indicated by different $NO_2$ VCDs at both sides of the gap (e.g. at the eastern side of the circle). Here it is important to note that even if similar $NO_2$ VCDs are measured at both sides of a gap, (like at the western side of the circle), gradients might still be present. Thus from the


differences of the NO$_2$ VCDs derived from the car MAX-DOAS measurements themselves only a lower limit of the errors caused by a gap can be estimated.

One simple way to estimate such errors is to perform the summation (eq. 2) in two directions and compare the corresponding results as in Shaiganfar et al. (2011). Since the values of the wind speed and direction in equation 2 are determined for the location of measurement i, but the distance $\Delta s_i$ is determined between measurement I and i+1, the direction for which the sum is calculated leads to a difference in the derived total NO$_2$ flux. For that reason we use the average NO$_x$ emissions from both directions for the determination of the NO$_x$ fluxes in our study. If no gaps are present (or if the NO$_2$ VCDs at both sides of a gap are similar) the results for both directions are almost the same. But for large gaps and large differences of the NO$_2$ VCDs at both sides of the gap the differences become large. For the measurements shown in Fig. 4 the difference of the results for both directions is 25%. Note that for most measurements the differences are much smaller. In section 4.6.1 we develop a more sophisticated method for the determination of the errors caused by gaps.

### 4.3 Partitioning correction

Since NO cannot be measured by car-MAX-DOAS measurements, but we are interested in the total NO$_x$ (NO + NO$_2$) emissions, the NO$_2$ fluxes derived from eq. 2 have to be multiplied by a partitioning correction factor:

$$F_{NO_x} = F_{NO_2} \cdot c_p \tag{3}$$

with

$$c_p = \frac{[NO_2] + [NO]}{[NO_2]} \tag{4}$$

The partitioning correction factor is typically between 1.1 and 2 and can e.g. be derived from model simulations (like in this study). If no model data are available, standard values for typical situations (e.g. Seinfeld and Pandis, 2012) can be used. The partitioning of NO and NO$_2$ depends mainly on the ozone concentration and the solar radiation. For high ozone concentration and low actinic flux a higher fraction of NO$_x$ is in the form of NO$_2$ (and vice versa). Figure 5 presents NO$_2$ VCDs derived from the car MAX-DOAS measurements (left) and partitioning ratios derived from simultaneous model simulations for 8 February 2010. Interestingly, the highest partitioning ratios are found at the same locations as the maximum NO$_2$ VCDs. In the following we always use the partitioning ratios from the model simulations at the locations of the maximum NO$_2$ VCDs for the conversion of the NO$_2$ fluxes into NO$_x$ emissions (eq. 3), because the derived NO$_x$ emission fluxes depend mainly on the difference between the maximum NO$_2$ VCDs (at the downwind side) and the background values.

Fig. A3 in the appendix presents the partitioning ratios derived from the model simulations at the locations of the maximum NO$_2$ VCDs for all days of both seasons. In summer on average smaller partitioning ratios (1.32) than in winter (1.51) are found, probably related to the higher ozone mixing ratios in summer (see Fig. A4). The results in Fig. A3 indicate a general problem: the deviations of the daily values from the seasonal average values are up to 30% and this rather large uncertainty directly propagates to the derived NO$_x$ emissions via equation 3.

One possibility to constrain the daily partitioning ratios might be to use the dependency on the wind speed (see Fig. 6). Here it should be noted that the partitioning factors are normalised (divided) by the seasonal averages in order to make the values for both seasons directly comparable. Decreasing (normalised) partitioning ratios (i.e. increasing relative contributions of NO$_2$ to the total NO$_x$) are found with increasing wind speed and vice versa. This finding is probably caused by a more effective turbulent mixing for days with higher wind speeds, which allows a more effective transport of ozone-rich air into





the air parcels with high NO concentrations. Indeed higher ozone concentrations and tropospheric column densities are found for higher wind speeds (see Figs. A4 and A5 in the appendix).

**4.4 Effect of NO$_x$ lifetime**

According to the generally short lifetime of NO$_x$ in the boundary layer, during the transport of the air masses from the
emission source to the location of the measurement part of the emitted NO$_x$ is destroyed. Thus the emissions derived from the measured NO$_2$ VCDs underestimate the true emissions. To correct for this underestimation, Shaiganfar et al. (2011) applied a so called lifetime correction factor:

$$c_\tau = e^{\frac{r}{v \cdot \tau}}$$

(5)

It is calculated from the wind speed v, the distance r between the city center and the location of the highest NO$_2$ VCDs and
the lifetime τ. For the measurements around Paris we assume a NO$_x$ lifetime of 3 hours and 12 hours in summer and winter, respectively (see e.g. Beirle et al., 2011). Here it should be noted that these lifetimes are rough assumptions, and on individual days large deviations from the assumed values might occur. But especially for wind speeds above about 2 m/s the effect of the limited lifetime of NO$_x$ and thus of the uncertainties of the assumed lifetimes are small (the correction factor is close to unity), Only in cases with low wind speeds larger effects occur (see Fig. 7). Here it is interesting to note that for our
measurements the effect of the wind speed dominates the variability of the lifetime correction factor, while the distance between the emission source and the measurements has only a small influence (Fig. A6 in the appendix). The errors of the derived NO$_x$ emissions caused by the uncertainties of the lifetime are discussed in more detail in section 4.6.4.

**4.4.1 Lifetime correction for the influx**

Shaiganfar et al. (2011) applied the lifetime correction only for the total emissions from the encircled area (difference
between influx and outflux). However, in cases with a high influx of NO$_x$, a lifetime correction should also be applied for the influx of NO$_x$, because only part of the NO$_x$ which is transported into the encircled area will actually reach the location of the outflux measurement. In such cases (see example in Fig. 8) the total flux will be underestimated if no lifetime correction for the influx was performed. In most cases, however, the influx of NO$_x$ is rather small and thus the neglect of the lifetime correction for the influx has only a small effect (a few percent) on the derived emissions.
We tested two versions of the lifetime correction for the influx. In the basic version a lifetime correction factor is determined in the same way as for the outflux and (its inverse) is applied to the derived NO$_x$ influx before it is subtracted from the outflux. In the more sophisticated version, the lifetime correction is only applied to the difference of the NO$_2$ VCDs with respect to the minimum NO$_2$ VCDs at the upwind side. This procedure takes into account that a large part of the observed NO$_2$ VCDs actually represents a homogenous background concentration, which is present at both the upwind and downwind
sides. This NO$_x$ background is probably mostly located in the free troposphere, where the atmospheric lifetime is longer than in the boundary layer. Thus the lifetime correction is only applied to the enhancements over this background.
To demonstrate the effect of the lifetime correction for the influx, we calculated the NO$_x$ emissions for 28 January 2010 (Fig. 8) in three ways:
a) without a lifetime correction for the influx: The resulting NO$_x$ emissions are $8.53 \cdot 10^{25}$ molec/s.
b) with the basic lifetime correction for the influx: The resulting NO$_x$ emissions are $9.68 \cdot 10^{25}$ molec/s.



c) with the sophisticated lifetime correction for the enhancements over the background: The resulting $NO_x$ emissions are 9.05 $\cdot$ $10^{25}$ molec/s.

In the following we apply the sophisticated version of the lifetime correction of the influx for all days. Here it should, however, be noted that the example shown in Fig. 8 is a rather extreme case and for most of the days the effect of the influx

correction is small (a few percent). However, it should also be noted that a lifetime correction for the influx is especially important for measurements with small differences between the outflux and influx.

### 4.5 Emission upscaling using nighttime lights

Like in Shaiganfar et a. (2011) the spatial distribution of nighttime lights (NOAA, National geophysical Data Center, 2006, http://www.ngdc.noaa.gov/dmsp/download_radcal.html, Ziskin et al., 2010) measured from satellite is used to upscale the

derived $NO_x$ emissions to a defined area around the city of interest. The corresponding distribution around Paris is shown in Fig. 9. The upscaling factor is defined as:

$$c_{upscaling} = \frac{L_{full\ area}}{L_{circle}} \tag{6}$$

Here $L_{full\ area}$ is the integral of the nighttime lights over the full area (latitude-longitude ranges as shown in Fig. 9), and $L_{circle}$ is the integral over the area inside the circle used for the car MAX-DOAS measurements. For most days during the

MEGAPOLI campaign the driving routes include large parts of the city, and the corresponding upscaling factors are between 1.23 and 2.11.

In addition to the upscaling factors calculated using the nighttime lights, we also calculated upscaling factors based on the distribution of $NO_x$ emissions in the emission inventory used for the model simulations (see Fig. 2). A scatter plot of both upscaling factors is shown in Fig. 10. The slope of the regression line and and the correlation coeffiient ($r^2$) are close tu unity,

but the scatter increases slightly with increasing upscaling factors. This finding is not unexpected taking into account the different quantities used for the upscaling (and also their different spatial resolution). Especially for small circles large deviations between the spatial distributions of nighttime lights and the $NO_x$ emissions are expected, e.g. due to the effect of strong and localised emission sources (e.g. power plants), which are not well represented by the nighttime lights.

### 4.6 Error estimation

In the previous sub-sections several steps for the calculation of $NO_x$ emissions from car MAX-DOAS observations were discussed. Each of these steps is subject to specific uncertainties. Some of these uncertainties are directly related to each other, e.g. the uncertainties of the wind speed and the lifetime correction. The following main uncertainties for the determination of the $NO_x$ emissions can be identified:

a) Sampling effects:

-measurement gaps

            -small circles causing large upscaling factors

            -time difference between measurements of influx and outflux

b) Measurement uncertainties:

            -errors of the derived $NO_2$ VCDs

c) Meteorological effects:

            -errors in residence time over the area, especially for low wind speeds





-changing wind speeds and wind directions

d) Chemical effects:

-uncertainties of the lifetime correction factor

-uncertainties of the partitioning factor

Some of these error sources can be minimised by an optimised planning of the driving routes. In particular, measurements should be performed around rather large circles, and bows close to the city center should be avoided. Other factors like the meteorology cannot be influenced by the experimentalists. But based on the meteorological conditions, decisions about when measurements are meaningful or not could be made (especially situations with very low wind speeds or highly varying wind fields should be avoided). While the lifetime correction factors can usually be calculated with low uncertainties (except for very low wind speeds), the uncertainties caused by the partitioning factor can be large. The effect of the time difference between the measurements of the influx and outflux can in general be neglected, because the temporal variability of the background is rather low. It has, however, to be taken into account that the derived $NO_x$ emissions are only representative for a specific time period of the day (mainly depending on wind speed and the diameter of the driving circle, see also section 5). In the following the different error sources are quantified in more detail.

### 4.6.1 Errors caused by gaps in measurements

Due to the finite number of measurements, the total flux of $NO_2$ has to be determined by a sum (eq. 2) instead of an integral (Eg. 1). I.e., at a given location $i$, the derived $VCD_i$ is applied along the complete distance $\Delta s_i$ towards the next location. In case of large distances $\Delta s_i$ ("gaps"), this procedure introduces uncertainty in the resulting emissions.

As shown in section 4.2 problems caused by large gaps can be identified in a simply way by comparing the emissions calculated in opposite directions along the driving route (eq. 2). In the following, we present a more sophisticated way for the quantification of the unceratinties related to gaps, based on error propagation.

In order to estimate the error of F due to gaps, we first estimate the uncertainty of VCD simply by the standard deviation of all measurements $VCD_i$:

$$\Delta VCD = std(VCD_i) \tag{7}$$

and thus the error of a single summand in Eq. 2 as

$$\Delta VCD_i \cdot \omega_i \cdot \sin(\beta_i) \cdot \Delta s_i \tag{8}$$

The error of the Flux is then given as

$$\Delta F = \sqrt{\sum_i \left(\Delta VCD_i \cdot \omega_i \cdot \sin(\beta_i) \cdot \Delta s_i\right)^2} = \Delta VCD \cdot \sqrt{\sum_i \left(\omega_i \cdot \sin(\beta_i) \cdot \Delta s_i\right)^2} \tag{9}$$

This approach assumes that the error of the single summands can be estimated from the statistical distribution of VCDs, assuming them to be independent. In reality, however,

1. VCDs for neighbouring locations are generally correlated, and

2. the variability of VCDs (and thus the potential error caused by gaps) is generally higher for the outflow than for the inflow.

To account for this, we modify the error estimate by

1. ignoring summands with $\Delta s_i <3$ km in eq. 9 (terms with sufficient spatial sampling should not contribute), and

2. calculating the uncertainty of in- and outflow separately (just defined by the sign of the individual summands). For the inflow, $\Delta VCD$, and thus $\Delta F$, is generally lower than for the outflow.





The total uncertainty is then just given by the root of sum of squares of the uncertainties of in- and outflow. This results in a realistic error estimate for errors introduced by gaps, as long as the std reflects the true uncertainty, i.e. the existing measurements actually reflect the variability of the $NO_2$ distribution.

### 4.6.2 Errors caused due to upscaling

As shown in section 4.5 the error of the upscaling factor increases with increasing upscaling factor (for small circles). We suggest to describe the error of the upscaling factor by the following empirical formula:

$$error_{upscaling} = \left(c_{upscaling} - 1\right) \cdot 0.4 \tag{10}$$

By this simple formula it is ensured that for a (hypothetic) driving route encircling the whole area (upscaling factor = 1) the uncertainty of the upscaling factor would be zero; and that for increasing upscaling factors also the uncertainties increase (up
to about 45% for the largest upscaling factor used in this study). In Fig. 10 it was shown that the deviation of the upscaling factors derived from the spatial distribution of the nightime lights or the emission inventory differ by about 10% for large upscaling factors. Thus our formula probably overestimates the uncertainties caused by the aplication of the scaling factor. Note that for other locations with different spatial patterns of the $NO_x$ emissions this simple parameterisation might not be appropriate.

### 4.6.3 Errors caused by the variability of the wind field

We quantify errors related to variations of the wind field by calculating the $NO_x$ emissions not only for the average values of the wind speed and wind direction (see section 4.1), but also for wind speeds changed by ±2m/s and wind directions changed by ±20°. Such variations are often found for measurements around full circles for the Paris measurements. Here it should be noted that the assumed variations of wind speed and direction also partly account for by uncertainties of the assumed $NO_x$
height profiles (see section 4.1), since wind speed and direction change with altitude.
In Fig. 11 two days with extremely variable wind fields are shown. On 24 January 2010 (left) the wind speed changed by more than a factor of two, on 29 January 2010 (right) the wind direction changed by more than 60°. For both days, large differences of the $NO_x$ emissions derived using either averaged or variable wind data were obtained (36% for 24 January; more than 100% for 29 January).
We also determined the $NO_x$ emissions using the spatio-temporally varying wind fields (interpolated to the exact locations and times of the individual car MAX-DOAS measurements) and compared them to the results derived from using the averaged wind data .
Fig. 12 displays the relative differences for all days versus the wind related errors calculated as described at the beginning of this sub section. For most days, the relative differences are small (<10%), but larger differences are found for days with large
errors caused by the variability of the wind fields (see also Fig. A7 in the appendix). This is an important finding because it indicates that the errors caused by the variability of the wind field are well represented by the simple approach to estimate the wind-related errors. Also, for days with small wind-related errors the averaged wind speed and direction can well be used without introducing significant additional errors. Here it should be noted that spatio-temporally resolved wind data are usually not available.



### 4.6.4 Errors caused by the lifetime correction

Similar to the errors caused by variations of the wind field also the errors caused by uncertainties of the lifetime correction are calculated. Here we assume variations of the lifetime by ±25%. Here it should be noted that this deviation should be seen as a rough estimate of the uncertainty of the lifetime, and on individual days the deviations from the assumed values might be
larger. However, as shown in section 4.6.6 the error caused by uncertainties of the lifetime are generally much smaller than other error sources. Thus our choice of the lifetime uncertainty is not critical.

### 4.6.5 Errors caused by the partitioning correction

In this study we derive the partitioning ratios from the model data. We find that the partitioning ratios depend systematically on season, but also vary from day to day (Fig. A3 in the appendix). The day to day variation probably mainly reflects
variations in meteorology and solar radiation, which affect the local partitioning ratios. For summer, an average partitioning ratio of 1.32 and for winter of 1.51 is found. These values might serve as first guess values also for other campaigns. A further refinement could be derived from the dependence of the partitioning ratio on wind speed (Fig. 6). But the validity of this dependence should be investigated in more detail in future studies.

Furthermore, it should be noted that close to strong emission sources (like power plants) only a limited fraction of the NO
might be already converted to $NO_2$ due to the titration of $O_3$. In such cases, the total $NO_x$ emissions will be systematically underestimated by our method. As discussed in Shaiganfar et al. (2011), the efficiency of the mixing of ozone-rich air with the $NO_x$ emission plume depends on the atmospheric stability and wind speed. As a rule of thumb, the distance of the car-MAX-DOAS measurements from strong emission sources should be about ≥5 km (see also Shaiganfar et al. 2011). In this study we use individual partitioning ratios derived from the model results (see section 4.3). The respective uncertainties are
difficult to quantify, and in the following we assume a value of 15%, which reflects the scatter of the daily values around the fitted regression line in Fig. 6 . If no model simulations are available, the corresponding uncertainties might be substantially larger, but part of the variability might be parameterised by the wind speed.

### 4.6.6 Total error

In Fig. 13 the errors discussed in this section are shown for all days of the MEGAPOLI campaigns. On most days the errors
caused by the variability of the wind field and due to gaps dominate the total uncertainties. Total errors range from ±30% to ±50%.

## 5 Consistency check based on CHIMERE model simulations

Based on the CHIMERE model simulations the consistency of CIM can be checked. For that purpose first the $NO_2$ VCDs for the exact locations and times of the car-MAX-DOAS measurements are extracted. Then the CIM is applied to the extracted
model $NO_2$ VCDs in the same way as for the measurements. The derived $NO_x$ emissions are compared to the TNO-MEGAPOLI emissions used as input for the model simulations (Fig. 2). Here the emissions for the individual time periods of the measurements are selected: the end of the time periods is the time of the observations of the maximum $NO_2$ VCDs. The begin of the time period is determined relative to the end time by subtracting the transport time of the air masses from the city center to the location of the maximum $NO_2$ VCDs. The corresponding time periods range from 2 to 4 hours.



The results for the consistency check are shown in Fig. 14. There the ratios of the derived emissions (CIM) versus the emissions (TNO) are displayed. Most of the ratios are close to unity, especially if their error bars (see section 4.6) are considered. However, for several days also large deviations (over- or underestimation) are found. For most of these days, also the errors are larger than on the other days.

It is interesting to relate the derived ratios to the contribution of specific problems affecting the CIM (see section 4.6). For that purpose in Fig. 14 different problems are indicated by different colours (for the criteria used for identification of the different problems see Table 1).

For days with no obvious problems (green dots) the ratios are in general close to unity and the error bars are rather small. The largest deviations from unity are found for days with large variability of the wind field. For days with small differences
between the outflux and the influx typically ratios below unity are derived, and for days with large lifetime corrections typically ratios larger than unity are obtained. For the other problems no clear systematic effects are found. Here it should be noted that the choice of the selection criteria for the different problems is somewhat arbitrary, but the selection criteria described in Table 2 might serve as a first orientation on whether a given measurement is suitable or not for the application of the CIM.

We further investigate possible reasons for the deviation of the ratios of the CIM results and the TNO-MEGAPOLI emissions from unity. For that purpose we display the ratios as function of the different quantities, which might affect the determination of the $NO_x$ emisions (Fig. A8 of the appendix). For most of these quantities no or only a weak correlation is found (especially for the winter data). For the summer data higher correlations, especially versus the lifetime correction factor and the partitioning ratio are found indicating a possible over- or under-correction of the respective influences. However, because
the correlations are still rather weak and are based only on few data points we did not made any change to our assumptions made for the lifetime correction (section 4.4).

Finally, we convert the emissions derived for different time periods of the day to the daily average values according to the respective diurnal variations of the emission input data (Fig. 2). The resulting daily average $NO_x$ emissions together with the $NO_x$ emissions derived for the periods of the measurements are shown in Fig. 15. The daily average values are in general
smaller than the emissions during the period of the measurements, because the measurements are always made during the day while the minimum of the emissions is found during night (Fig. 2). Here it is interesting to note that the ratio of the average summer values to the average winter values (0.78) is very close to the same ratio in the emission inventory for Paris (0.81, see also Fig. 2 top). Interestingly, on most weekend days (Saturdays: 18 July, 17 January, 13 February; Sundays: 31 January) not the lowest emissions are found. We have no clear explanation for this finding, but for the winter results it might be partly
related to variations of the domestic combustion processes. Note that in Fig. 15 only results for days with small uncertainties and small differences between CIM results for CHIMERE and TNO emissions are shown (13 days from 31 days with uncertainties of the car-MAX-DOAS measurements >100% and with ratios of the CIM results for CHIMERE and TNO emissions above 1.7 and below 0.6 are skipped).

## 6 Application to measurement data

In Fig. 16 the $NO_x$ emissions and the associated errors derived from the car MAX-DOAS measurements for all days during both campaigns are shown. In contrast to the results from the model simulations (Fig. 14) the errors include also the uncertainties of the determination of the $NO_2$ VCD (20%). Thus the error bars are in general larger than those in Fig. 14. Potential problems for individual days are indicated by the colours of the data points.



Like for the results derived from the model simulations, also for the car MAX-DOAS the smallest errors are in general found for days without obvious problems. However, the variability of the derived $NO_x$ emissions is larger than for the results derived from the model simulations indicating that the real variability of the emissions is probably larger than that of the model results. However, it should also be noted that most of the results could be reconciled taken the error bars into account.

In Fig. 17 the daily average emissions derived from car MAX-DOAS measurements are compared to those derived from the model simulations and to the TNO emissions.

In general higher $NO_x$ emissions are derived from the car-MAX-DOAS measurements than from the CHIMERE model simulations (and also compared to the TNO emissions). The differences between the car MAX-DOAS and CHIMERE results are higher in winter (about a factor of 2) than in summer (about a factor of 1.4). We have no clear explanation for these

differences, but it is interesting to note that they are similar to those found for the direct comparison of the $NO_2$ VCDs of car-MAX-DOAS measurements and CHIMERE model results for the same data sets (Shaiganfar et al., 2015). Interestingly, the highest emissions derived from car-MAX-DOAS in winter were found for days with wind from north-north-west.

## 7 Conclusions and perspectives

We applied the closed integral method (CIM) for the determination of the $NO_x$ emissions from Paris based on a large set of

car MAX-DOAS measurements during two measurement campaigns in summer 2009 and winter 2009/2010. The campaigns were organised in the frame of the European project MEGAPOLI (Baklanov et al., 2010; see also http://megapoli.dmi.dk/). The CIM is applied to car MAX-DOAS measurements made at large circles around Paris with diameters of about 20 to 40 km. On 9 days in summer 2009 and 22 days in winter 2009/2010 meaningful emission estimates were possible. Compared to previous applications of the CIM, the large number of measurements during the MEGAPOLI campaign allowed to

investigate important aspects of the CIM. In particular the applicability of the CIM under various atmospheric conditions could be tested. Another important advantage of the MEGAPOLI campaigns is that simultaneous atmospheric model simulations of the CHIMERE model with high spatial resolution (3 x 3 km²) were available for all days. Based on these model data it was in particular possible to test the consistency of the CIM. For that purpose first the model data are sampled at exactly the same locations and times of the car MAX-DOAS measurements. Then the CIM is applied to the extracted

model results and the corresponding $NO_x$ emissions are determined. Finally the derived emissions are compared to the input emissions used in the model simulations. From this consistency check important information about favorable or non-favorable conditions for the application of the CIM were derived. In most cases, the errors caused by uncertainties and the variability of the wind fields contribute most to the total error budget. From this finding we conclude that in particular situations with low wind speeds and/or large variability of the wind directions should be avoided. Also gaps and uncertainties

of the partitioning ratio are important error sources. Based on the consistency check, we also deduced a set of criteria on whether measurement conditions are suitable or not for the application of the CIM. We also discuss the individual steps of the CIM, in particular the effect of lifetime correction (for the influx and outflux) and the correction for the partitioning of NO and $NO_2$, and develop methods to calculate the error budget of the derived $NO_x$ emissions. From the consistency check based on the CHIMERE model we find that the derived total errors are consistent with the deviations between the emissions

derived from the application of the CIM to the model data and the input emissions (TNO-MEGAPOLI). Typical errors are between ±30% and ±50%.

We apply the CIM to car MAX-DOAS observations for summer and winter. In summer daily average $NO_x$ emissions of $4.2 \cdot 10^{25}$ molecules/s, and in winter daily average $NO_x$ emissions of $7.8 \cdot 10^{25}$ molecules/s are derived for Paris. These value are by a factor of about 1.4 and 2.0 larger than the corresponding input emissions (and also the emissions derived from the





application of the CIM to the model data) in summer and winter, respectively(for many days these deviations are larger than the error bars). Similar ratios are also found for the comarison with the TNO-MACC-III inventory (1.5 and 2.3 for summer and winter, respectively). These findings, are in contradiction with previous comparison studies based on ground based (Zhang et al., 2013) and aircraft measurements (Petetin et al., 2015), which found that the model simulations systematically

overestimate the measurements for July 2009. The reason for the systematic discrepancies with our results is not clear. The most probable reason is that our measurements are sensitive for the integrated $NO_2$ concentration in about the lowest 3 km of the atmosphere, while the prevous studies compared in situ observations at the ground or between about 500m and 900m altitude. Here it is interesting to note that similar ratios between the emissions derived from the car-MAX-DOAS data and CHIMERE results were also found for the direct comparison of the $NO_2$ VCDs derived from car-MAX-DOAS or CHIMERE

(Shaiganfar et al., 2015). Another interesting finding is the enhanced seasonal cycle and the larger day to day variability of the $NO_x$ emissions derived from the car-MAX-DOAS measurements compared to that of the input emissions (see Fig. 2). Currently we have no explanation for these findings. While we have explored the uncertainties associated with the car-MAX-DOAS measurements in this paper, it should be acknowledged that exact emission timing per hour or per day in the emission inventories is also rather poorly defined. The time profile (Fig. 2) is an approximation but the same profile applies for every

15 year regardless of exact meteorological or traffic congestion conditions which may vary. Moreover, our results indicate consistently (but not always significant) higher emissions than the inventories and while it is difficult to extrapolate these to a yearly total, the idea that NOx emissions from road transport may still be underestimated is widespread. A high day to day variability was also found by Petetin et al. (2015). Results of the car MAX-DOAS measurements and model results for all days together with the wind fields are shown in Fig. A9 in the appendix.

**Acknowledgements**

The research leading to these results has received funding from the European Union's Seventh Framework Programme FP/2007-2011 within the project "MEGAPOLI", grant agreement n°212520. We are very thankful to several drivers, who supported the car-MAX-DOAS measurements: Thierry Marbach, Tobias Tröndle, Steffen Dörner, Christoph Hörmann, Bastian Jäcker, Sven Krautwurst. The authors would like to acknowledge J.C. Dupont and the SIRTA team for providing the

25 wind cube lidar data used in this study. We acknowledge Airparif (see http://www.airparif.asso.fr/en/index/index) for providing the emission data for the greater Paris region.

**Tables**

Table 1 Criteria used for the identification of different problems.

| Problem | Criterium |
|---|---|
| Large wind variability | Relative deviation of wind speed > 30%; deviation of wind direction >30° |
| Large lifetime correction | Lifetime correction factor > 1.5 |
| Gap / route close to the center | Gaps >14km, or Ratio of $NO_2$ flux left / right between 0.80 and 1.20 |
| Small difference between influx and outflux | Ratio outflux/influx ($NO_2$) <1.3 |
| Large partitioning ratio | Relative difference to seasonal average value larger than ±25% |





Table 2 Overview of days with several problems

| Day | Model simulations | MAX-DOAS measurements |
|---|---|---|
| 18.07.2009 |  | Large gap<br>Large difference left / right |
| 16.01.2010 | Large difference left / right<br>Deviating partitioning ratio | Large difference left / right<br>Deviating partitioning ratio |
| 17.01.2010 | Large gap<br>Deviating partitioning ratio | Large gap<br>Deviating partitioning ratio |
| 27.01.2010 | Large wind variability<br>Large difference left / right<br>Small difference out – in | Large wind variability<br>Large difference left / right<br>Small difference out – in |
| 28.01.2010 | Large gap<br>Small difference out – in<br>Deviating partitioning ratio | Large gap<br>Deviating partitioning ratio |
| 29.01.2010 | Deviating partitioning ratio<br>Large wind variability | Large difference left / right<br>Deviating partitioning ratio<br>Large wind variability |
| 01.02.2010 |  | Large difference left / right<br>Deviating partitioning ratio |
| 14.02.2010 | Large difference left / right<br>Large wind variability | Large difference left / right<br>Large wind variability |

Table 3 Comparison of the daily average $NO_x$ emissions from TNO with those derived from CHIMERE ($CIM_{CHIMERE}$) and car MAX-DOAS ($CIM_{MAXDOAS}$) for days with errors of the car-MAX-DOAS data <100% and differences of the CIM results for CHIMERE and TNO emissions <70%. For the calculation of the average the daily values are weighted by their individual errors. The values in brackets are calculated for all days. Emissions are given as $10^{25}$ molecules $NO_x$ per second.

| Season | TNO | $CIM_{CHIMERE}$ | $CIM_{MAXDOAS}$ | Ratio $CIM_{MAXDOAS}$ / $CIM_{CHIMERE}$ |
|---|---|---|---|---|
| summer | 2.9 (2.9) | 3.0 (3.0) | 4.2 (3.8) | 1.40 (1.27) |
| winter | 3.8 (3.6) | 3.9 (3.9) | 7.8 (6.1) | 2.00 (1.56) |

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

**Figures**

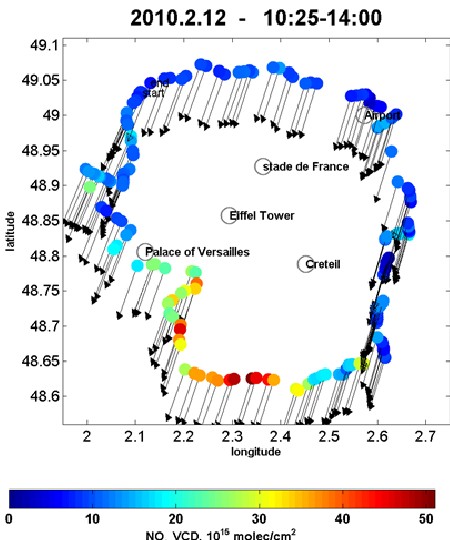

Fig. 1 Tropospheric vertical column densities (VCD) of NO₂ derived from car MAX-DOAS measurements around Paris on
12 February 2010 (each dot indicates an individual measurement). The arrows indicate wind speed and direction taken from the regional CHIMERE model. The average wind speed was about 8.5 m/s.





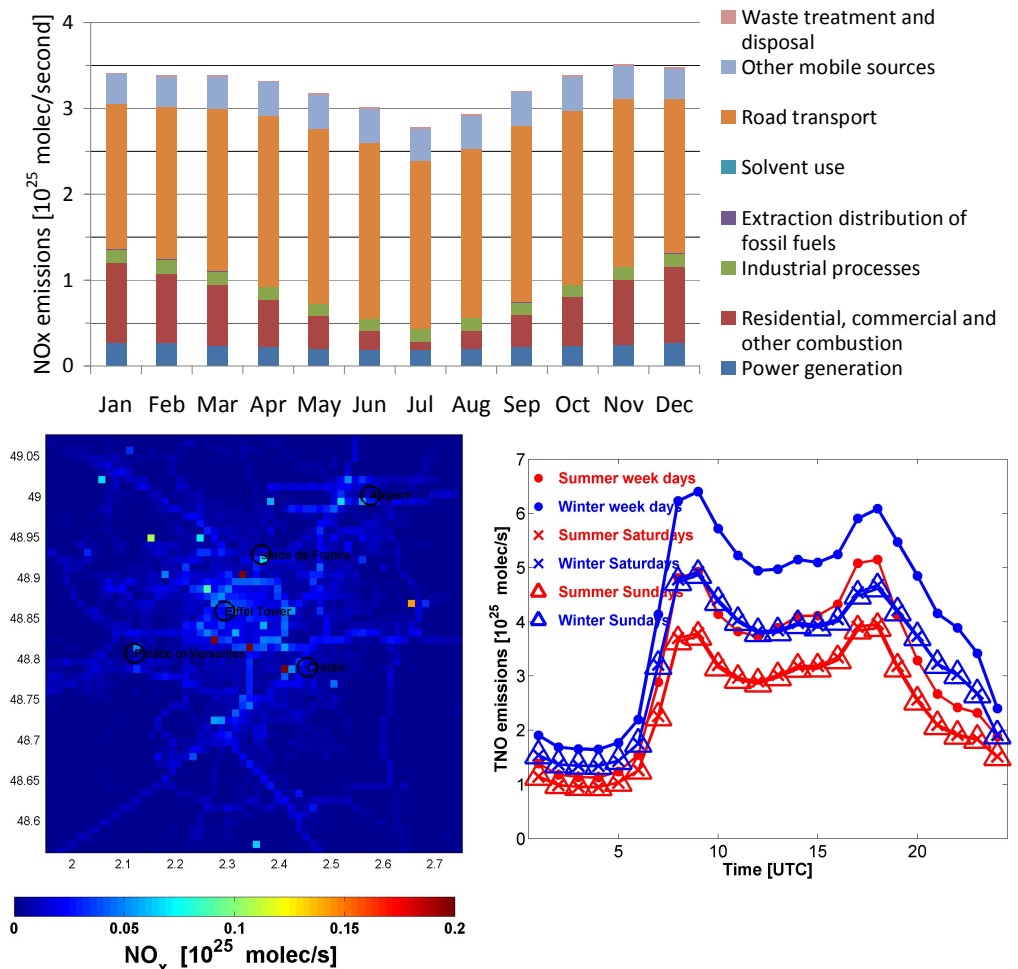

Fig. 2 Top: Seasonal cycle of the $NO_x$ emissions for the greater Paris area (Ile de France) in 2009 by source sector per month. Bottom: Examples of the spatial distribution (left) and the diurnal variation (right) of the integrated $NO_x$ emissions over the area shown in the left part. For the Paris region the emission data are available at 1 km resolution, but for the CHIMERE simulations they were averaged to the spatial resolution of the model (3km).



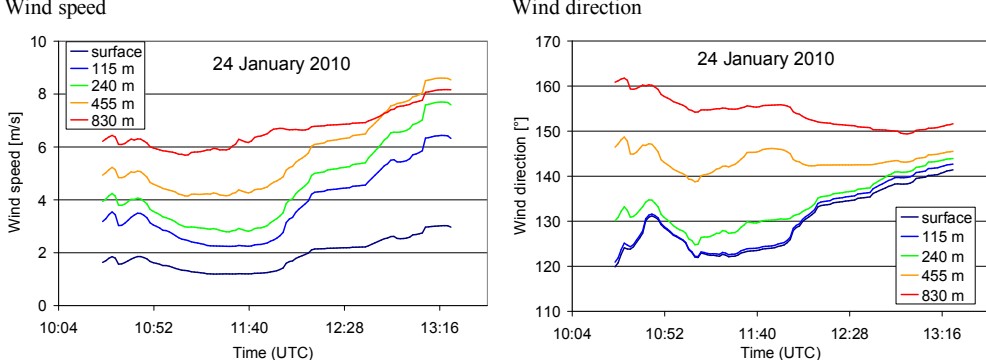

Fig. 3 Wind speed (left) and direction with respect to North (right) at different altitudes for 24 January 2010 obtained from the MM5 model.





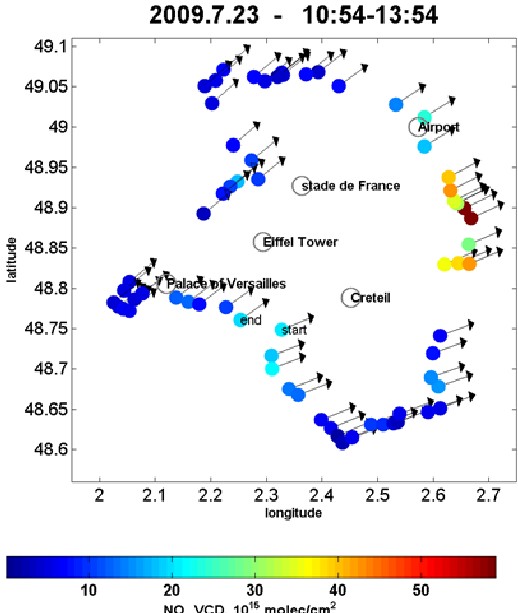

Fig. 4 Car-MAX-DOAS results for 23 July 2009. On this day several gaps due to instrumental problems occurred.





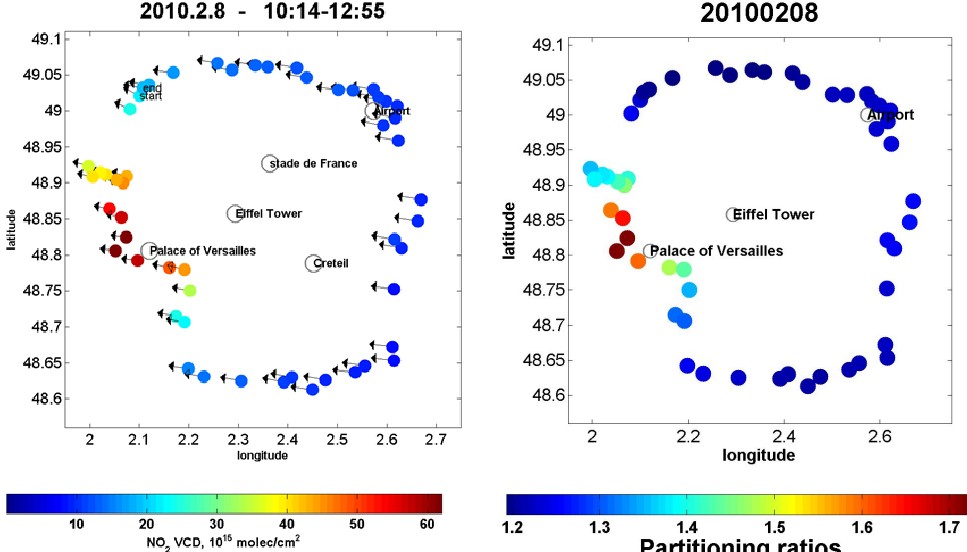

Fig 5 Left: NO$_2$ VCDs retrieved from car MAX-DOAS measurements on 8 February 2010. Right: partitioning ratios (NO$_x$/NO$_2$) derived from model simulations at the locations of the car MAX-DOAS observations. High partitioning ratios are found at the same locations of the maximum NO$_2$ VCDs.





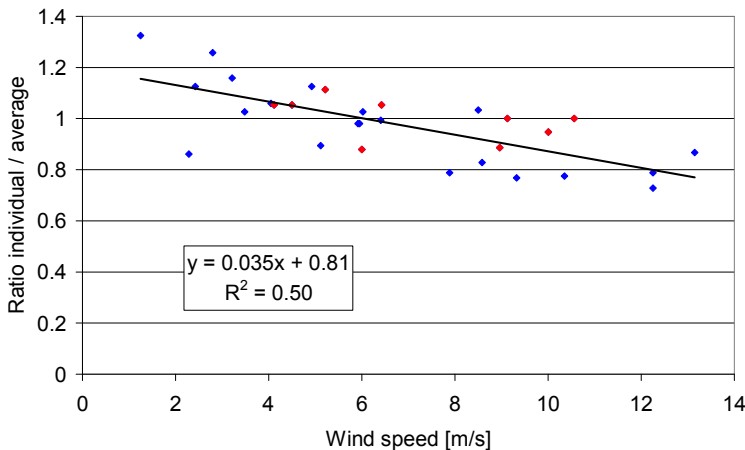

Fig. 6 Dependence of the normalised partitioning ratios on wind speed (the individual ratios are divided by the seasonal average). Red points represent summer and blue points winter data.





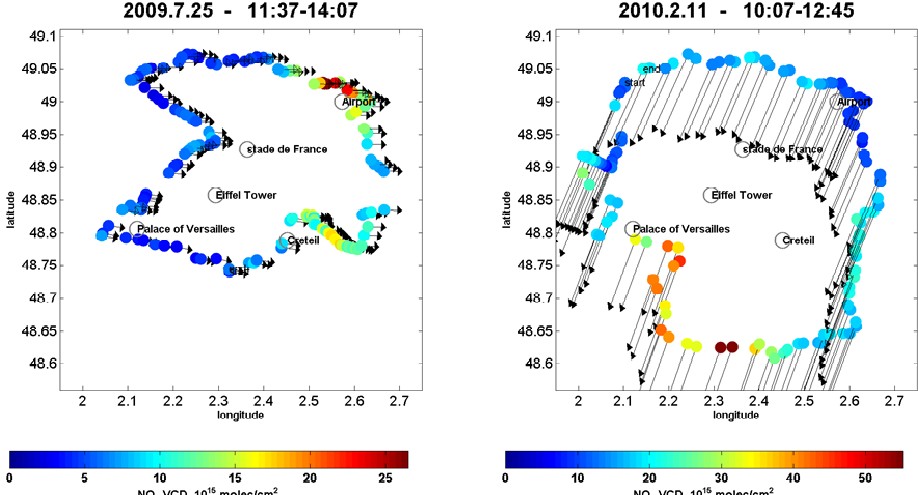

Fig. 7 Measured $NO_2$ VCDs and corresponding wind vectors for two selected days with extreme lifetime correction factors. For the day with low windspeed (left) a high lifetime correction factor of 1.71, and for the day with high windspeed (right) a low lifetime correction factor of 1.05 is found.





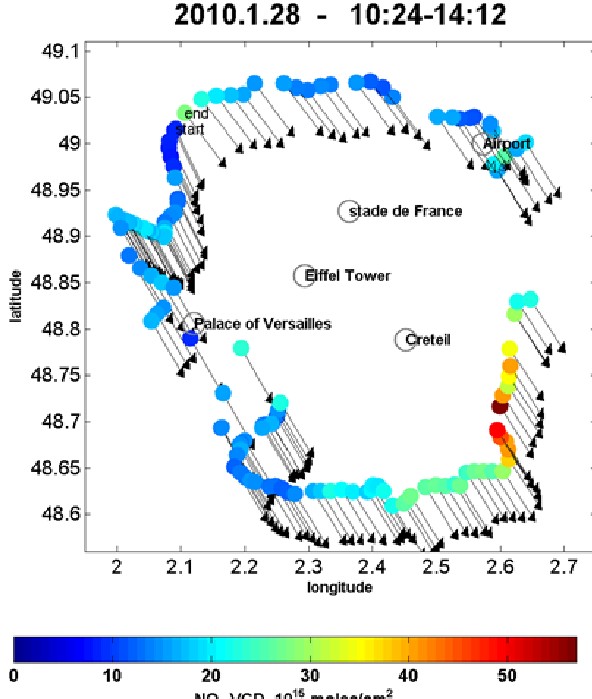

Fig. 8 NO$_2$ VCDs measured on 28 January 2010. On this day locally enhanced NO$_2$ VCDs are found at the upwind side of the circle.



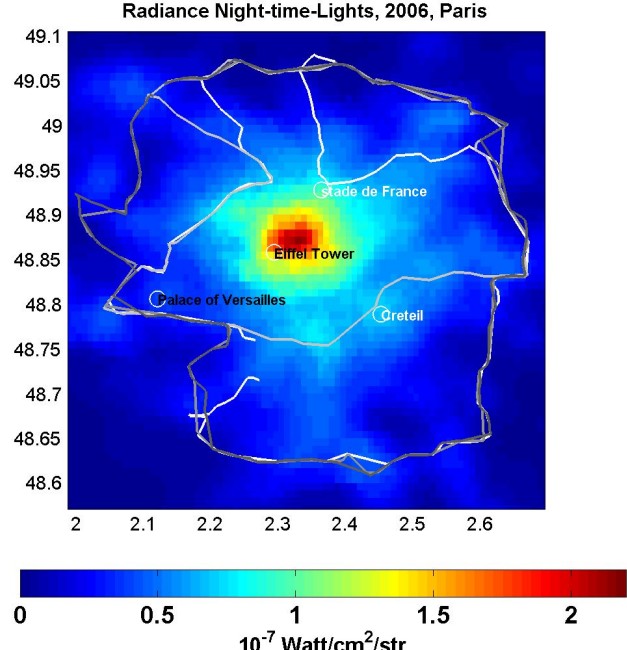

Fig. 9 Spatial distribution of the nighttime lights for the selected area around Paris. Different driving routes for 2 days in summer and 2 days in winter are shown.





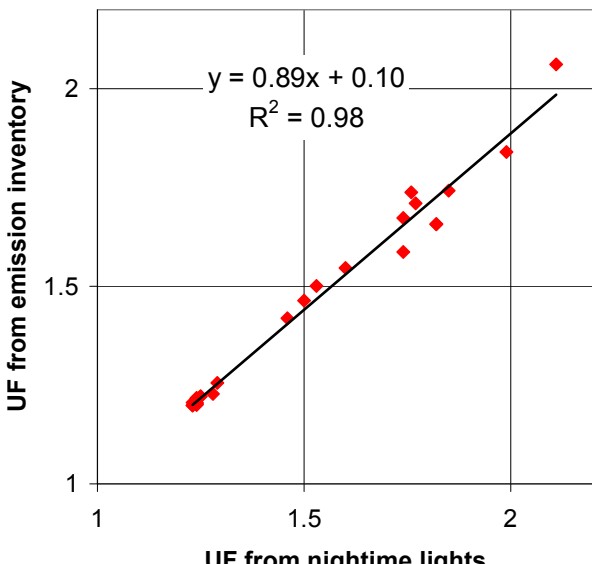

Fig. 10 Upscaling factors (UF) based on the distribution of the $NO_x$ emissions are plotted as function of the upscaling factor based on nighttime lights.




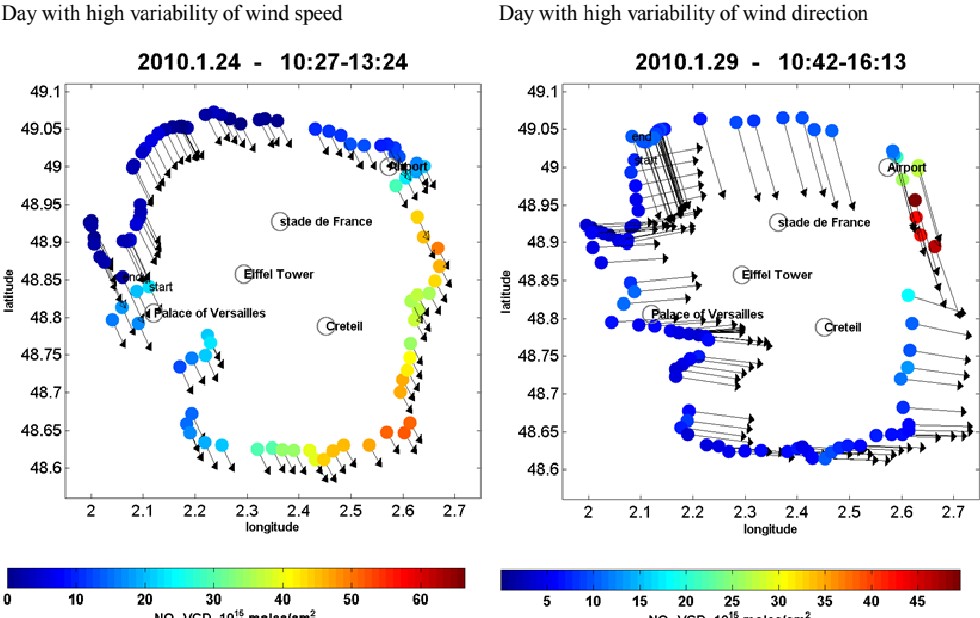

Fig. 11 Measured $NO_2$ VCDs and corresponding wind vectors for two selected days with extremely variable wind fields. On 24 Jaunary 2010 (left) the wind speed and on 29 January 2010 the wind direction showed large variations.





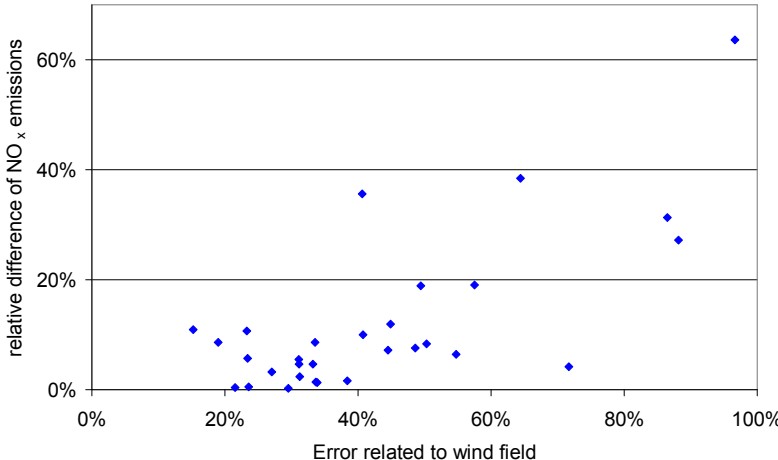

Fig. 12 Difference of the $NO_x$ emissions derived from either averaged or variable wind data plotted versus the wind-related error calculated as decsribed in section 4.6.3.





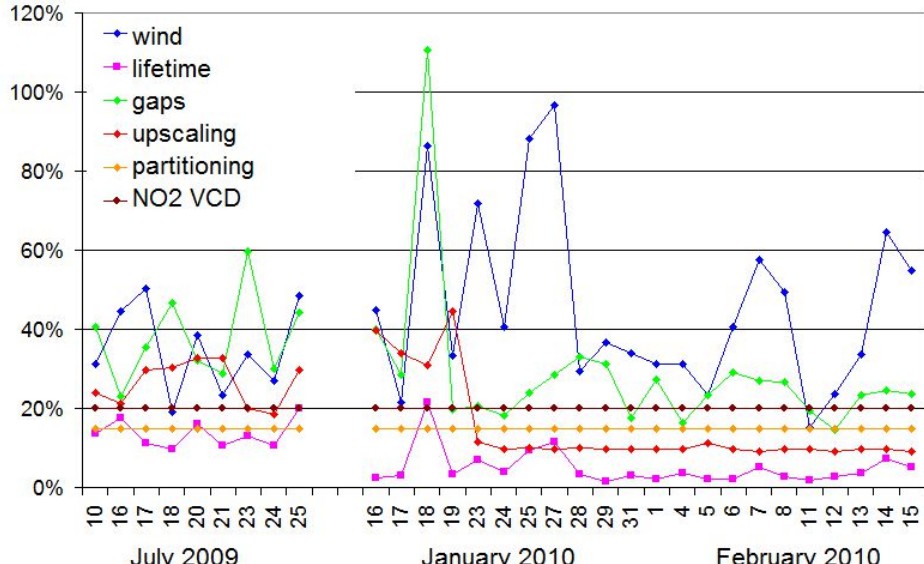

Fig. 13 Individual errors derived for all car-MAX-DOAS measurements considered in this study.




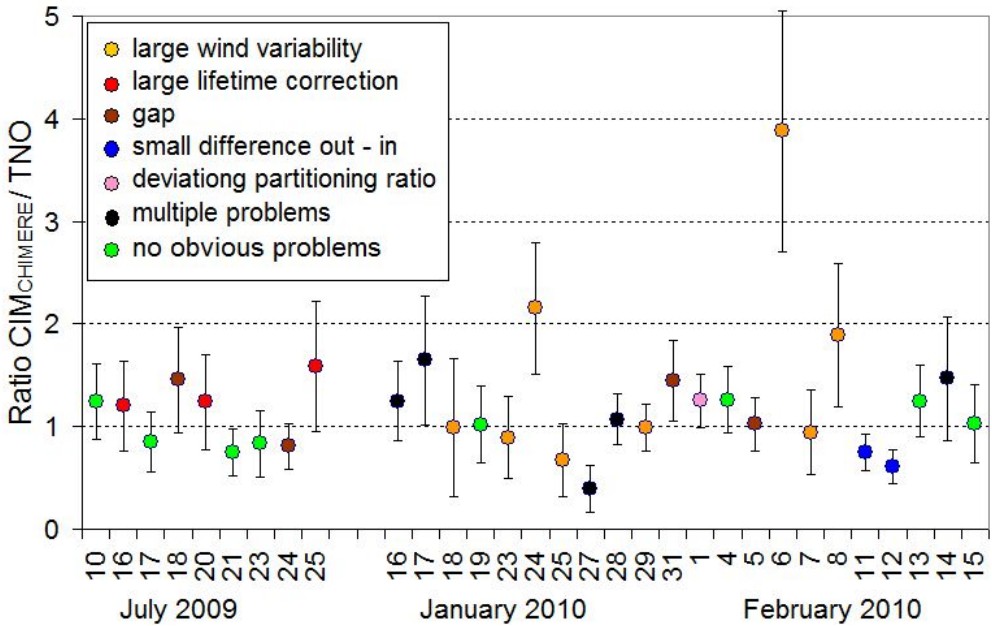

Fig. 14 Ratios of the derived $NO_x$ emissions ($CIM_{CHIMERE}$) and the TNO emissions used for the model simulations. For the CIM individual partitioning ratios for the different days are used (see section 4.3). The TNO emissions are averaged for the time period of the respective measurements (see text). The colours indicate different kinds of problems of the individual days (see also Table 2).



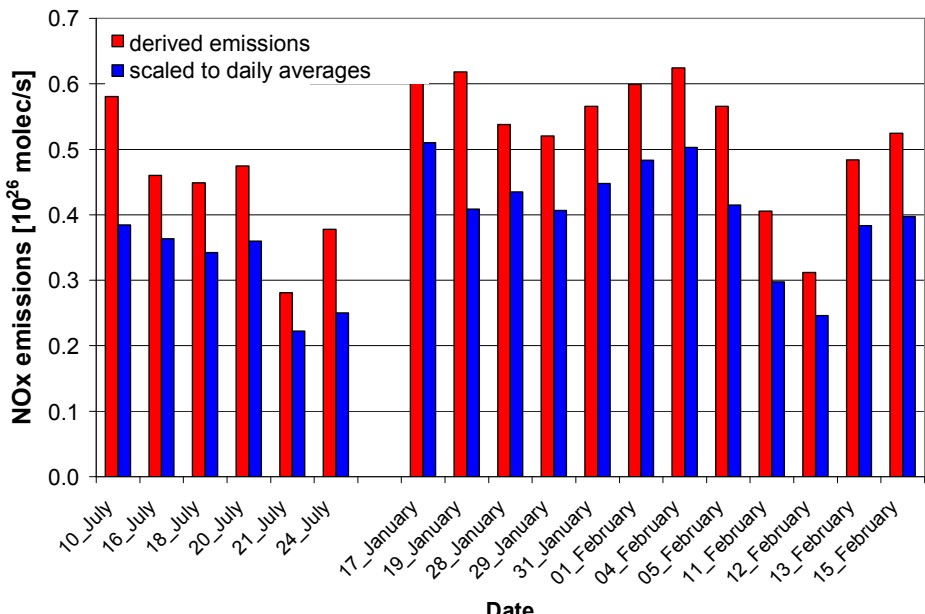

Fig. 15 Comparison of the derived emissions from CHIMERE model simulations for either the periods of the individual measurements or scaled to daily averages. Days with uncertainties of the car-MAX-DOAS measurements >100% and with ratios of the the CIM results for CHIMERE and TNO emissions above 1.7 and below 0.6 are skipped.





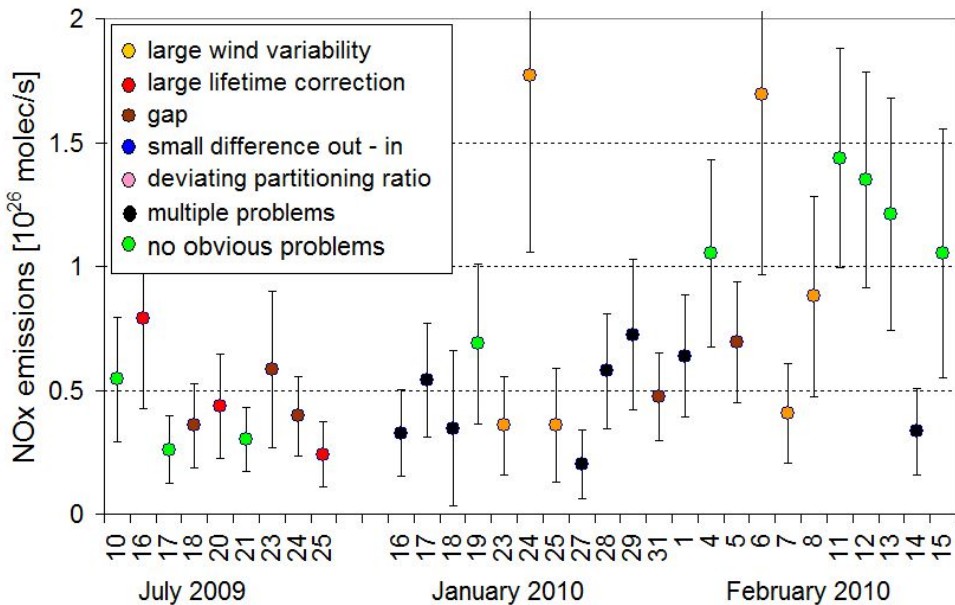

Fig. 16 NO$_x$ Emissions derived from the car MAX-DOAS measurements (using individual partitioning ratios and scaled to daily average values). The colours of the data points indicate different error sources (see also Table 2).





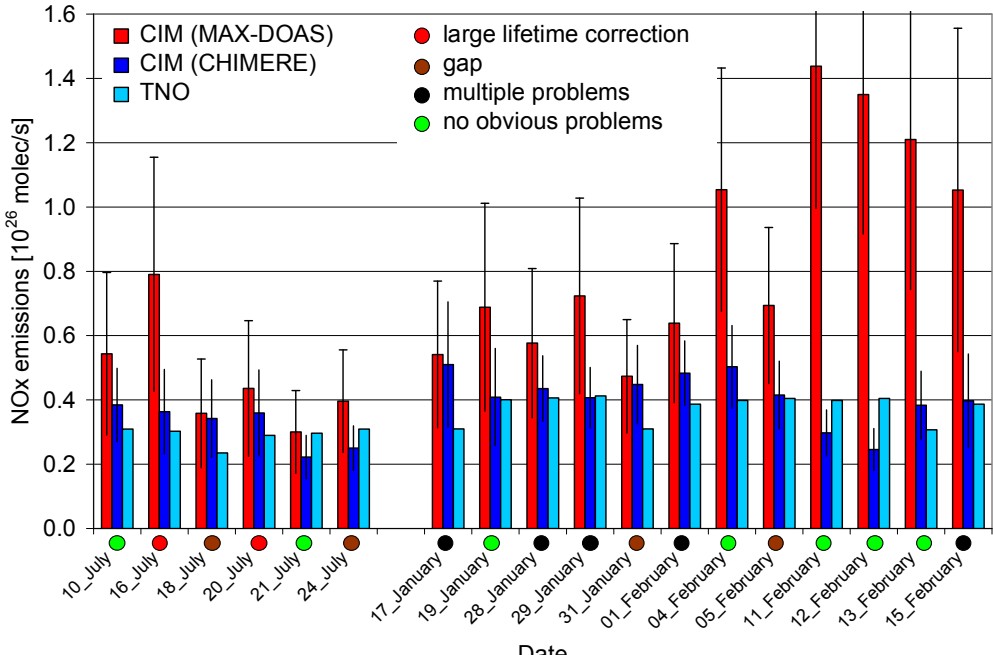

Fig. 17 Comparison of the NO$_x$ emissions derived from the car MAX-DOAS measurements (red) with from the CHIMERE model simulations (blue) and the corresponding TNO input emissions (light blue). All data represent daily average emissions. The NO$_x$ emissions from MAX-DOAS and CHIMERE were derived using daily partitioning ratios. Results are only shown for 'good' measurement days (days with uncertainties of the car-MAX-DOAS measurements >100% and with ratios of the CIM results for CHIMERE and TNO emissions above 1.7 and below 0.6 are skipped). The coloured disks below the bars indicate potential problems of specific days (same scheme as in Fig. 16).





**Appendix**

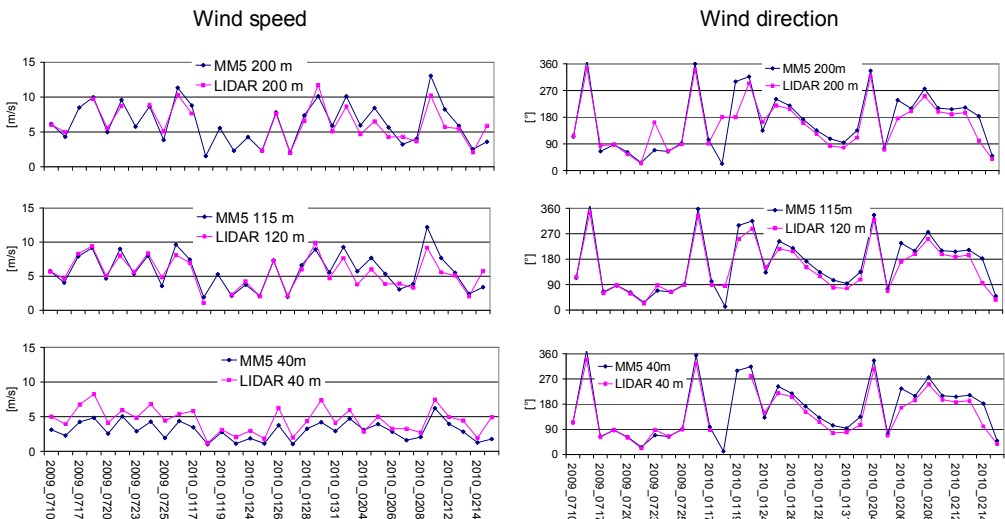

Fig. A1 Comparison of the wind fields (left: speed, right: direction) derived from the MM5 model and the wind LIDAR at the SIRTA site at Palaiseu in the South-West of Paris for three altitude levels.

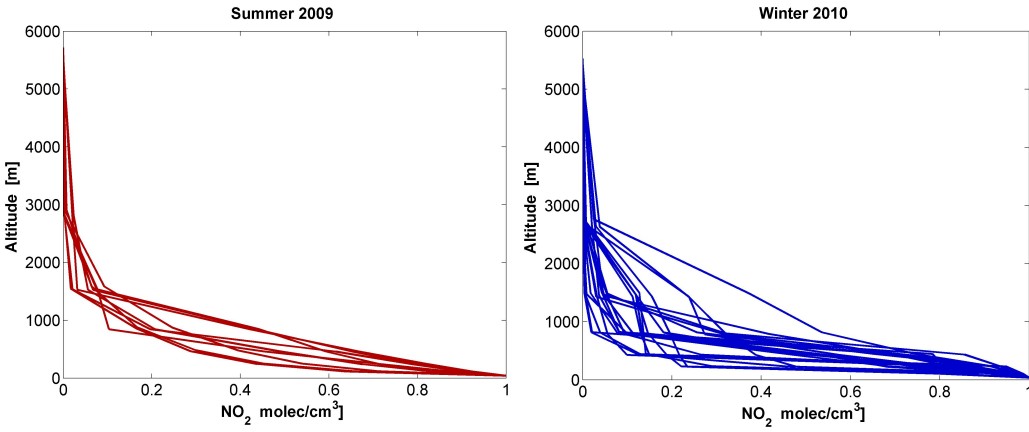

Fig. A2 Daily CHIMERE $NO_2$ profiles (normalised by the surface values) for Summer (left) and winter (right) extracted for the locations where the maximum $NO_2$ VCDs were measured by car-MAX-DOAS.





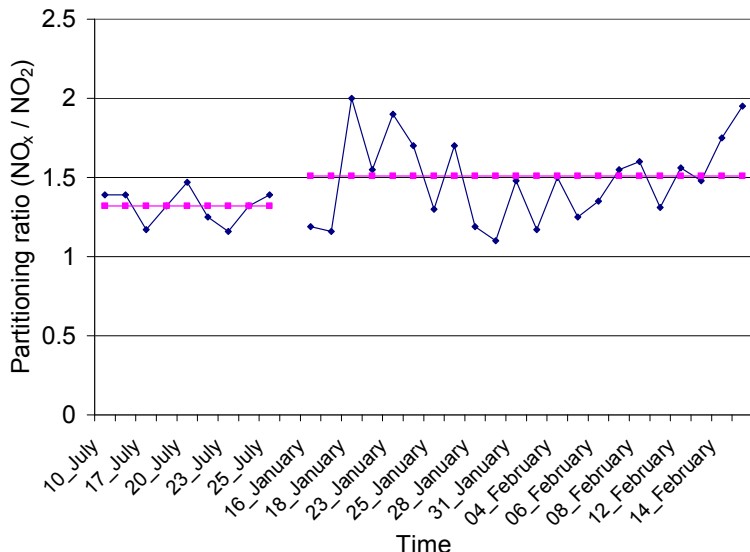

Fig. A3 Partitioning ratios derived from the model simulations for the locations of the maximum $NO_2$ VCDs for all days of the campaigns. A large day to day fluctuation, but also a systematic difference between winter and summer is found. The average ratios for summer and winter are 1.32 and 1.51, respectively.




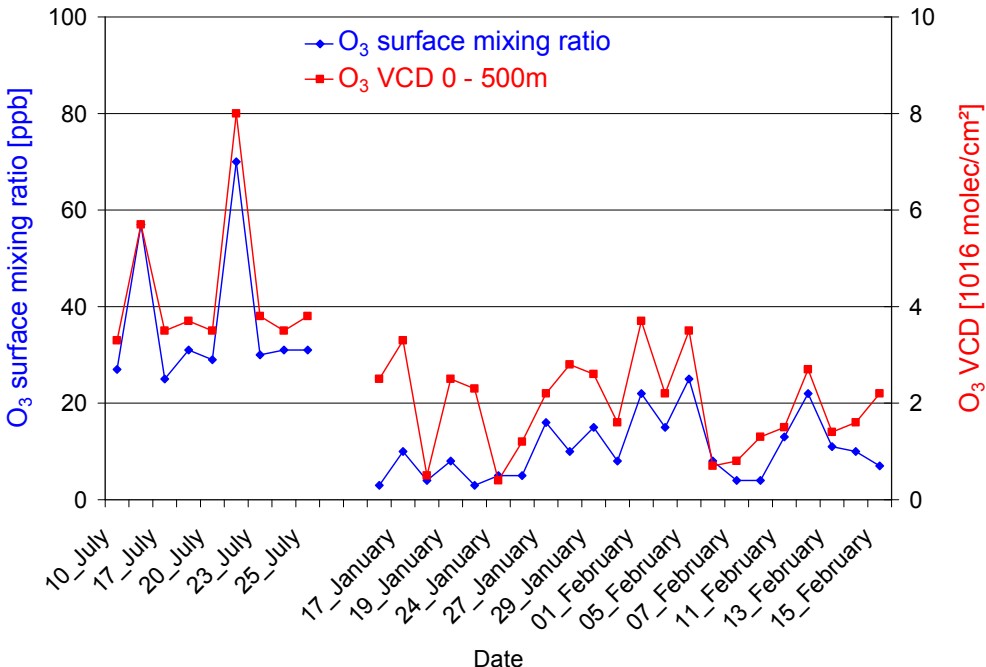

Fig. A4 $O_3$ mixing ratios and tropospheric VCDs at the location of the highest $NO_2$ VCDs for all days.

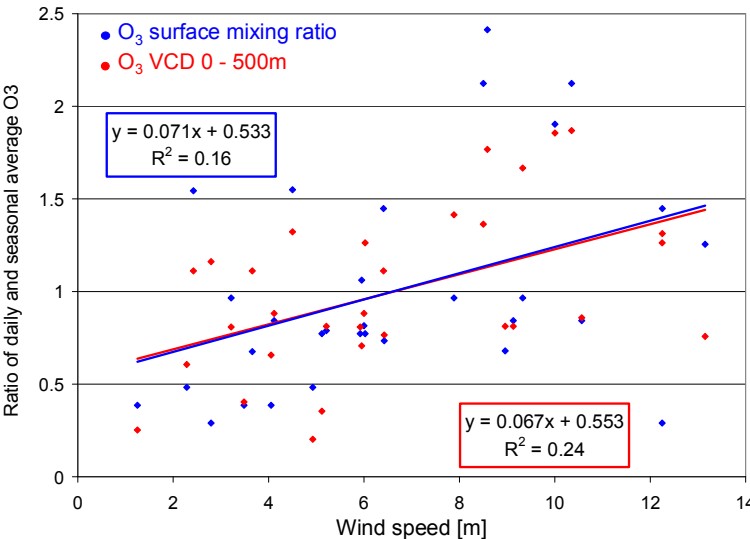

5    Fig. A5 Dependence of the surface mixing ratios and $O_3$ VCDs on wind speed. To account for the seasonal differences, the daily values are divided by the respective seasonal averages





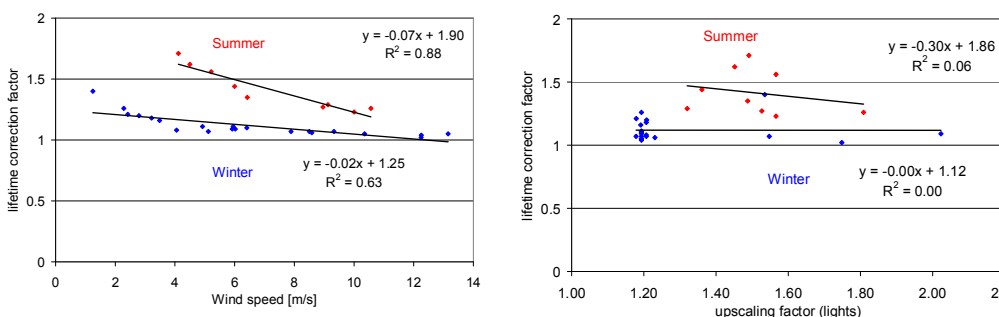

Fig. A6 Dependence of the lifetime correction factor on wind speed (left) and the size of the circle (right, represented by the upcaling factor).

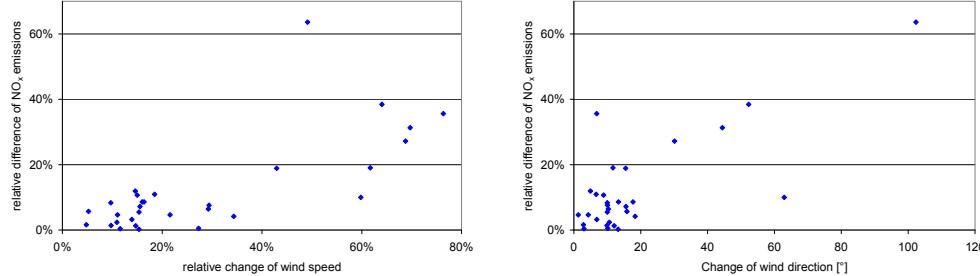

Fig. A7 Difference of the $NO_x$ emissions derived with either averaged or variable wind data plotted versus the variabilty of
10    the wind speed (left) and wind direction (right).




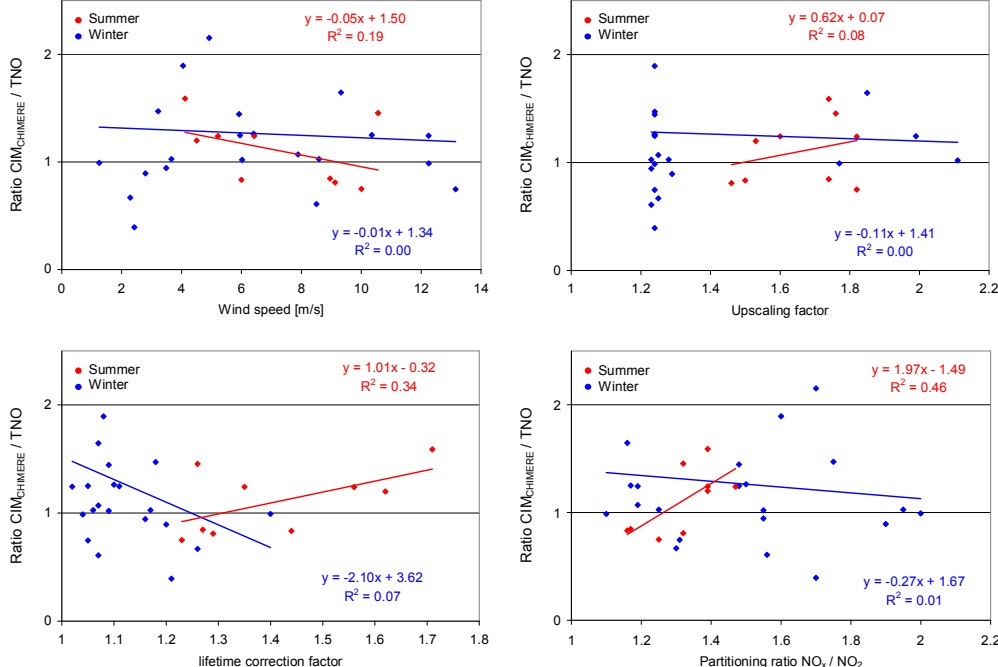

Fig. A8 Dependence of the ratio of the derived emissions (CIM$_{CHIMERE}$) and TNO NO$_x$ emissions on different parameters for summer (red) and winter (blue).




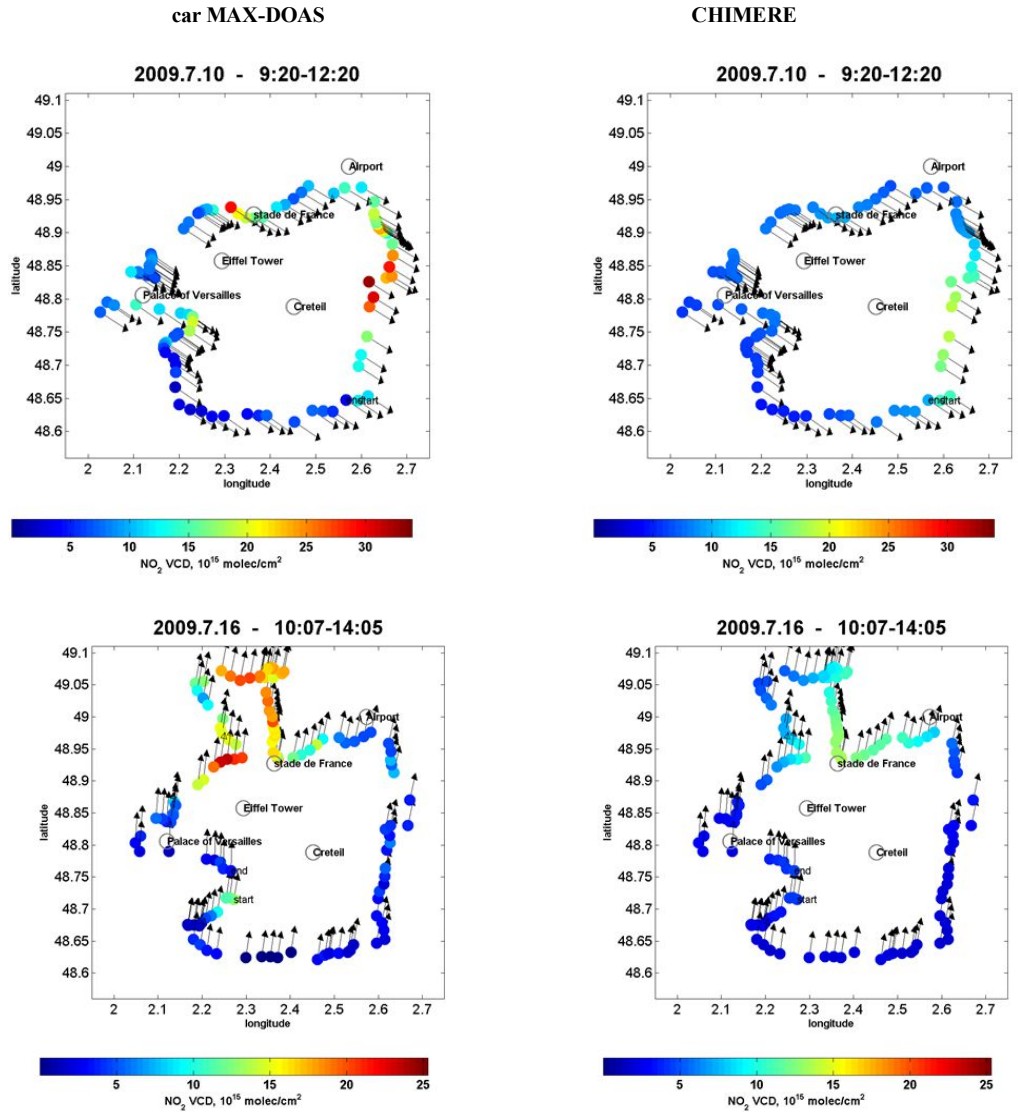





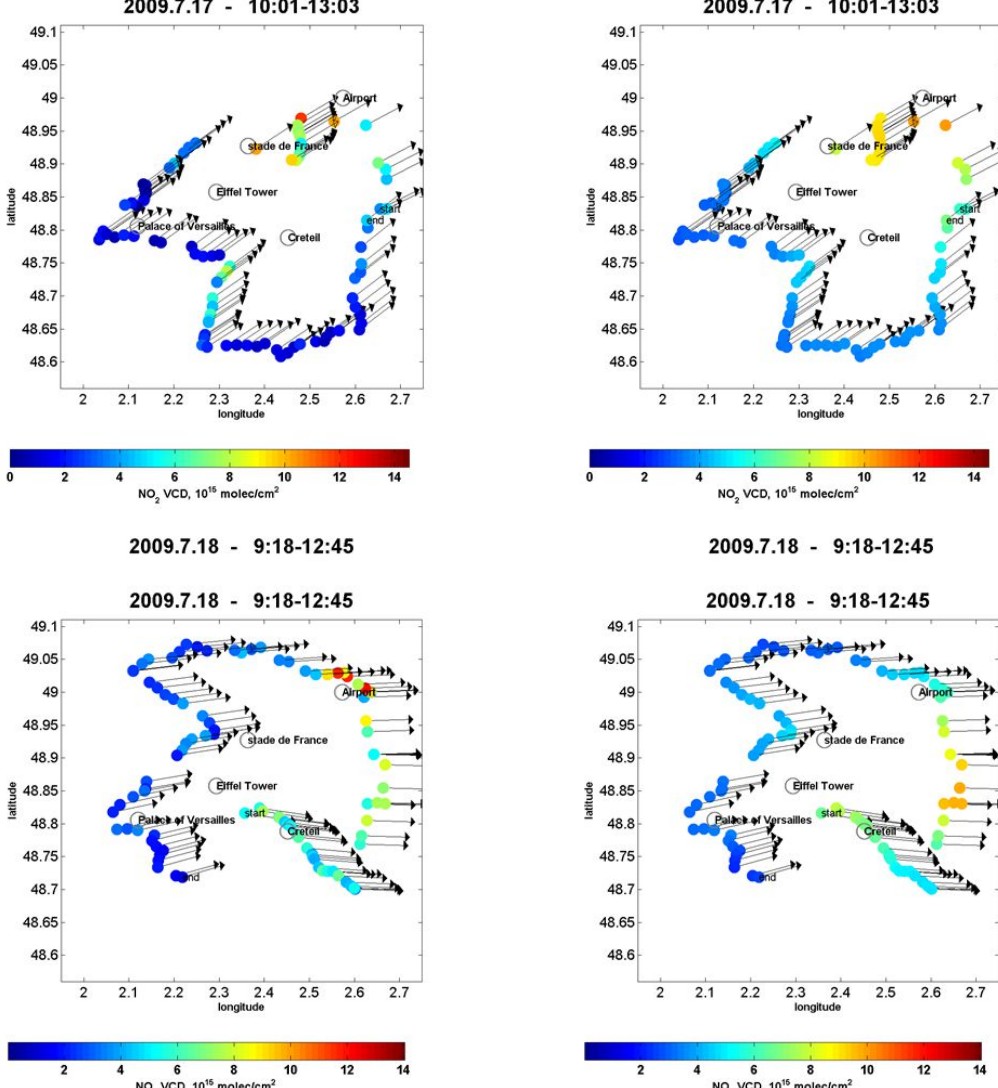





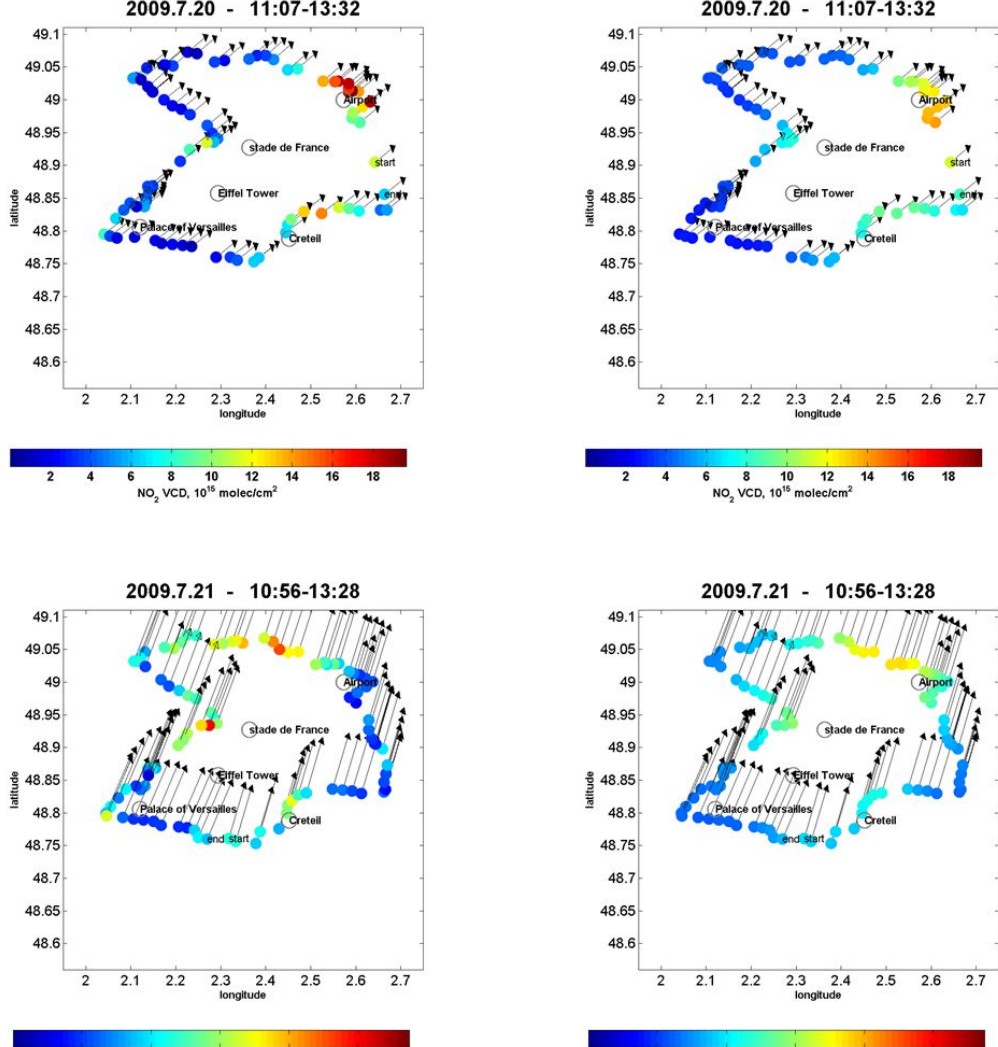



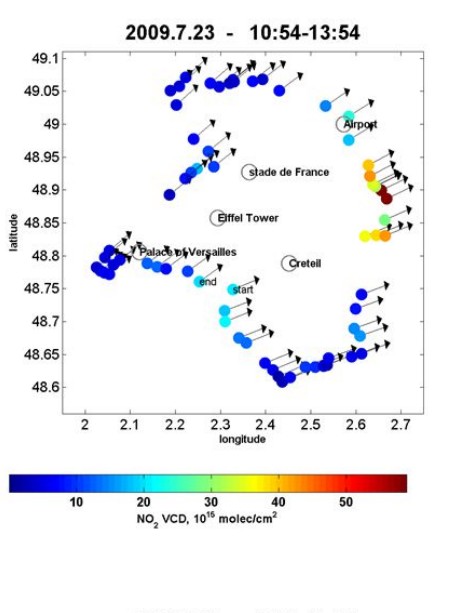

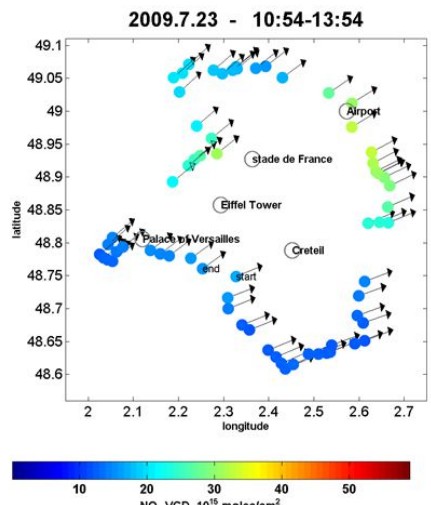

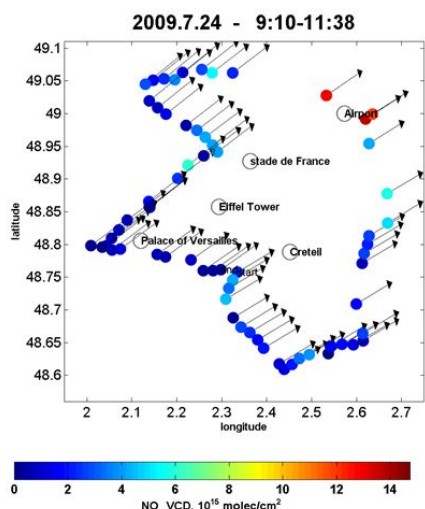

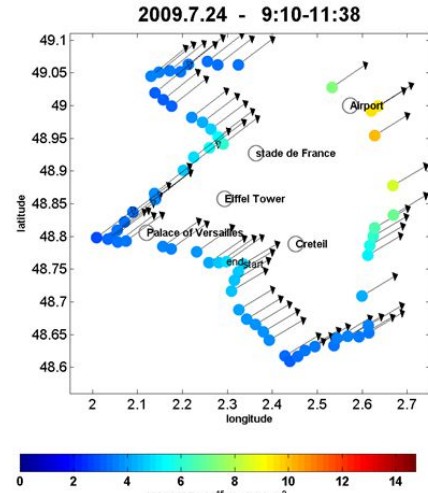




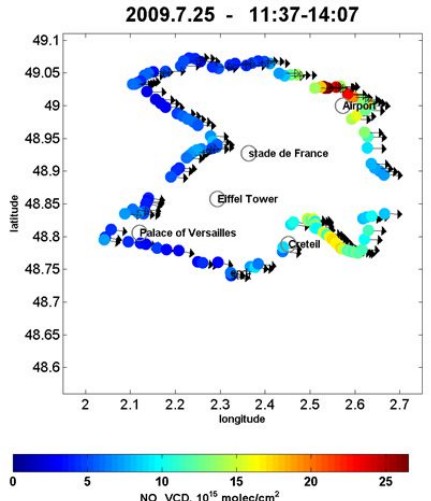
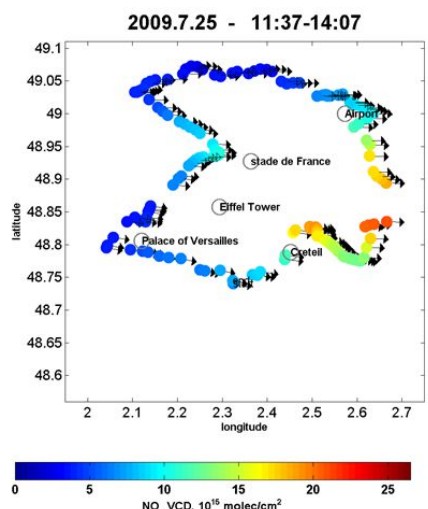

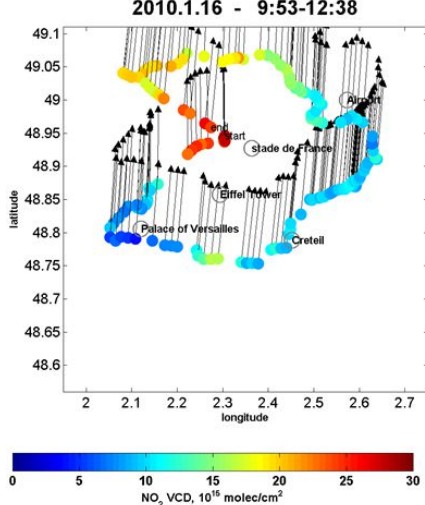
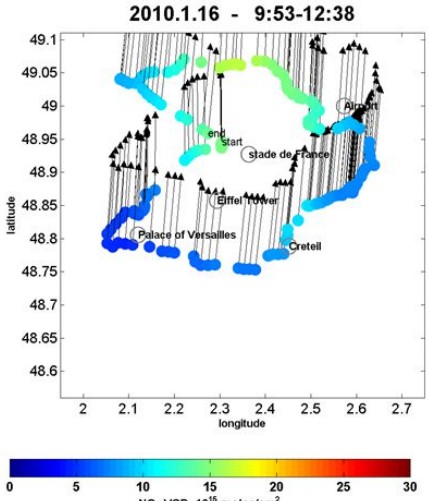





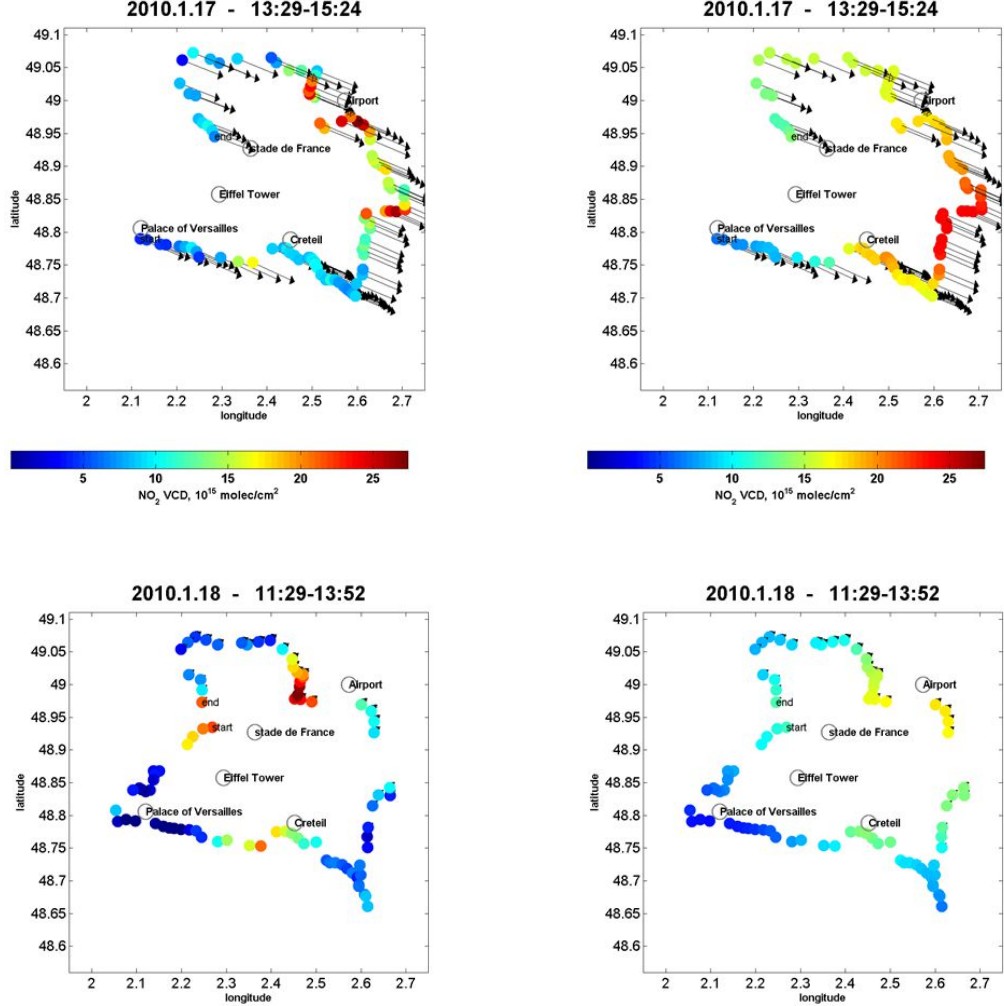



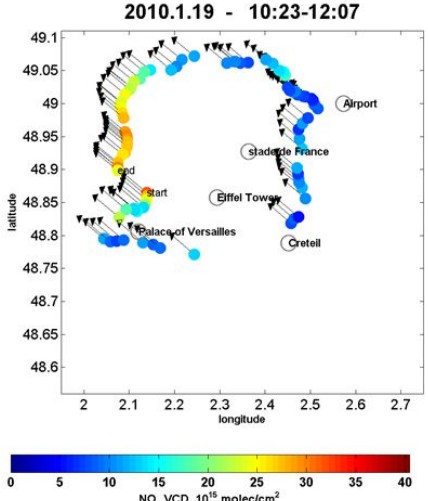
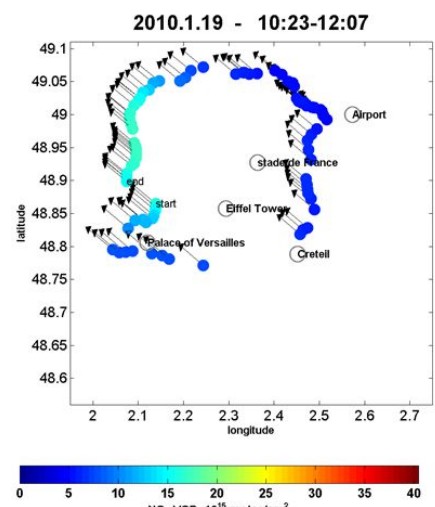

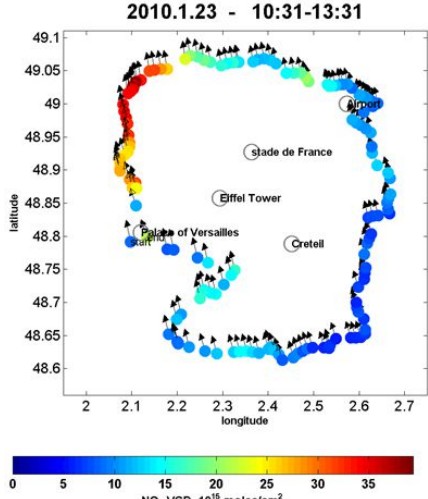
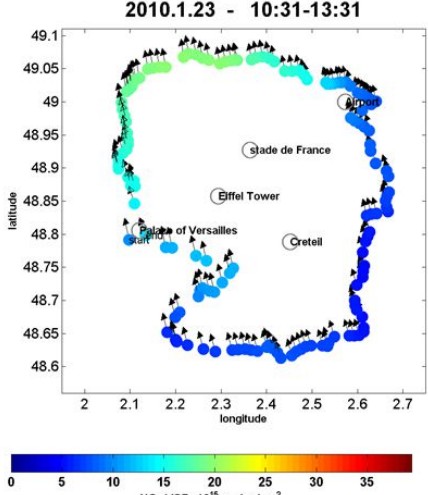



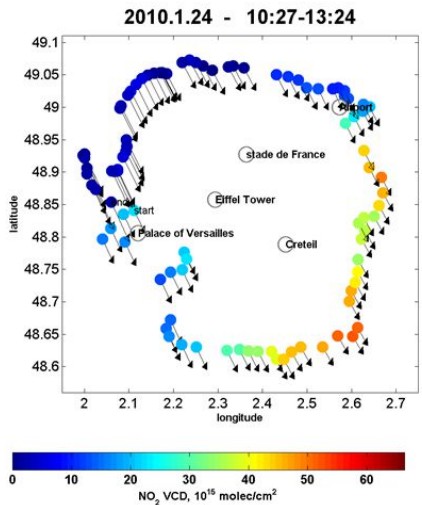

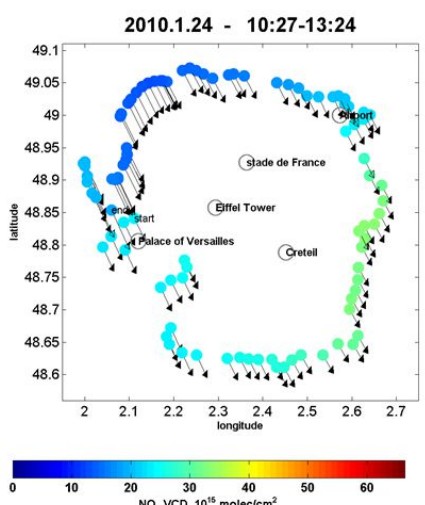

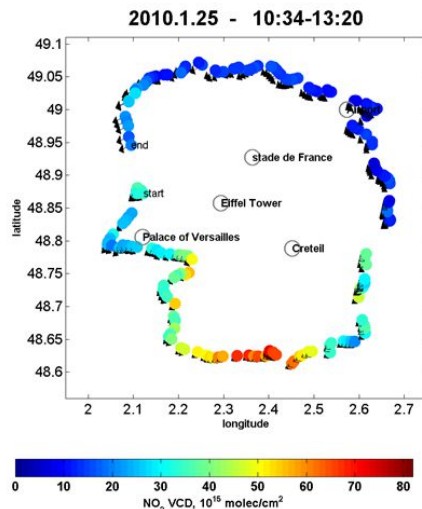

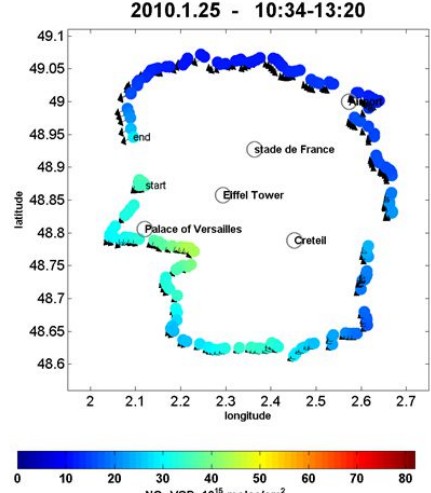





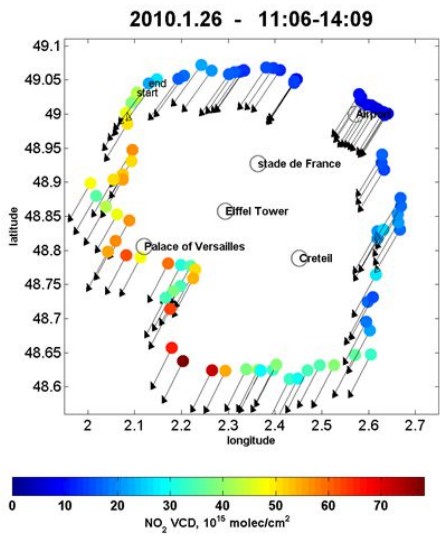

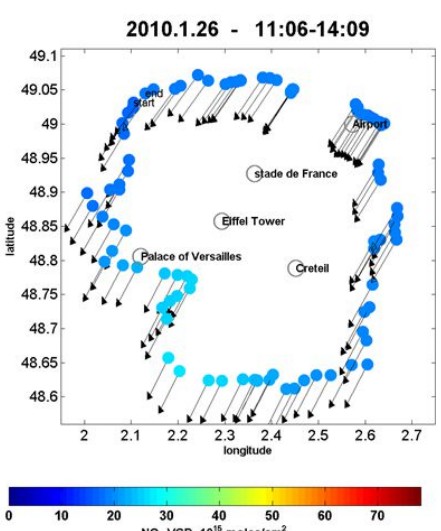

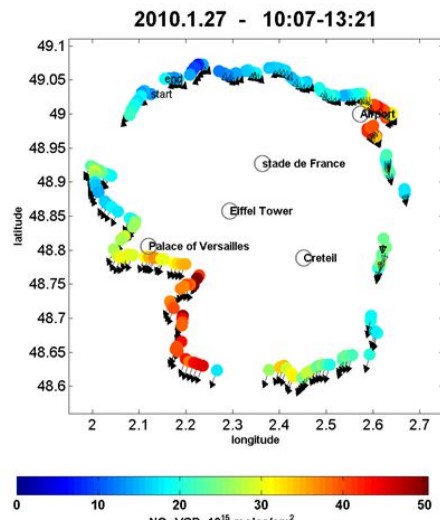

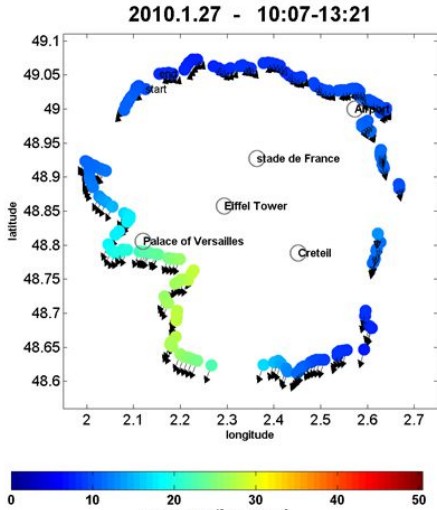

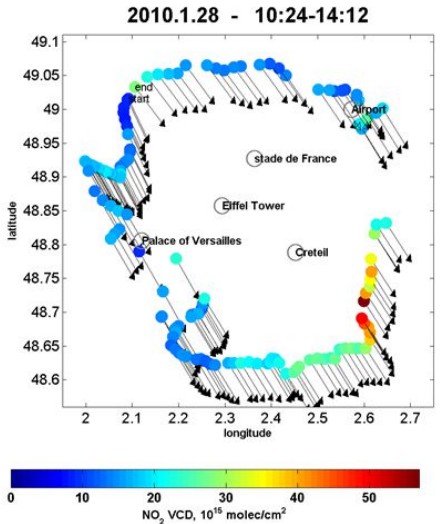

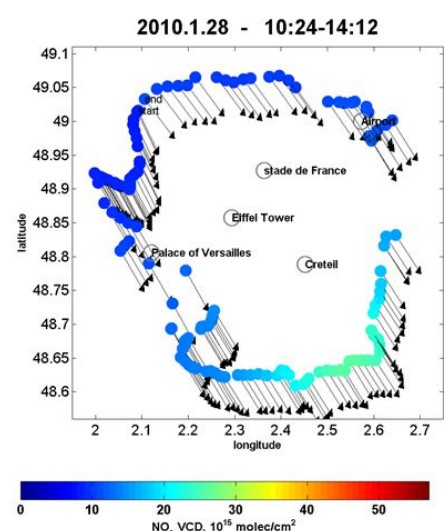

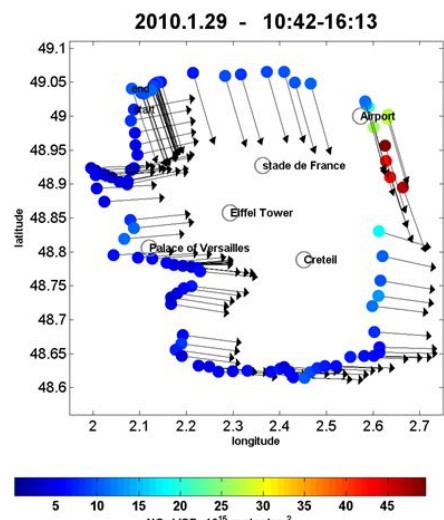

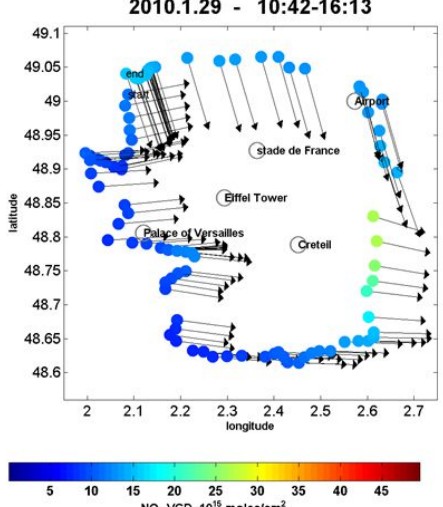




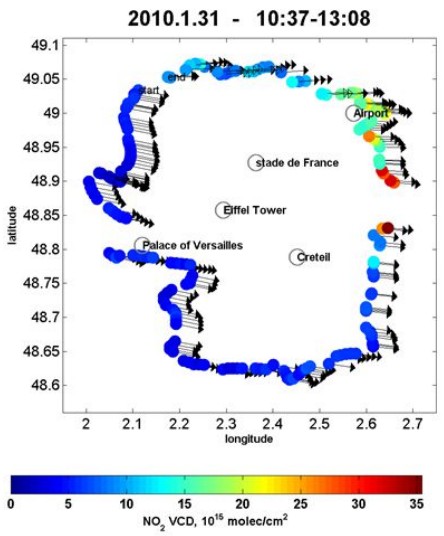
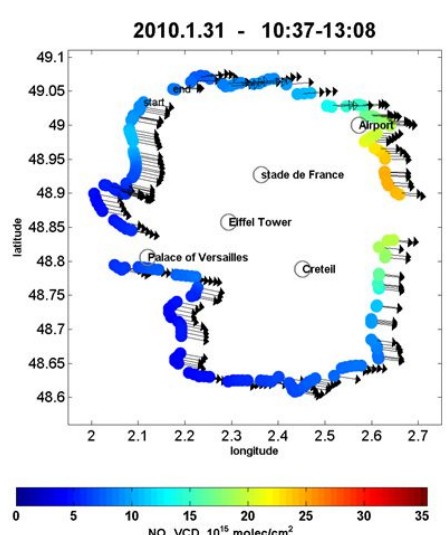

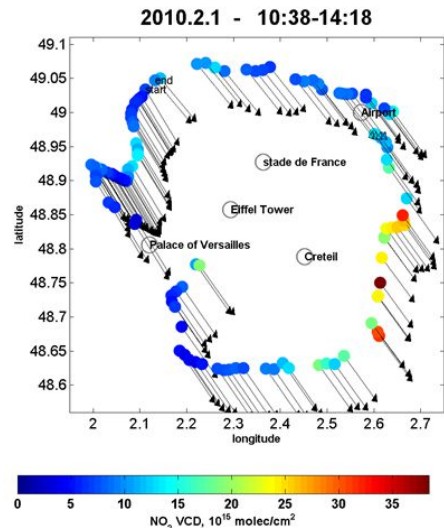
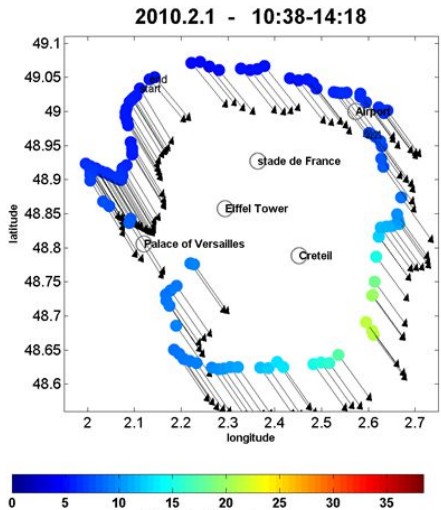





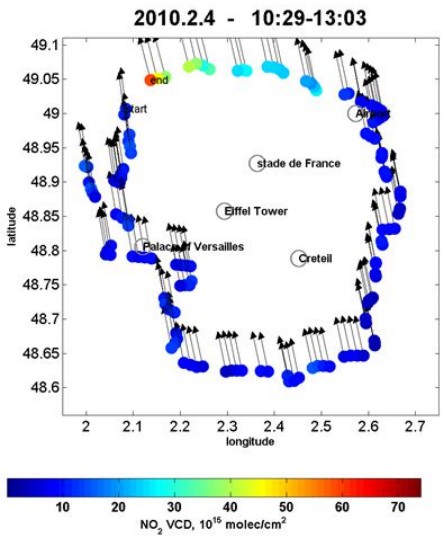

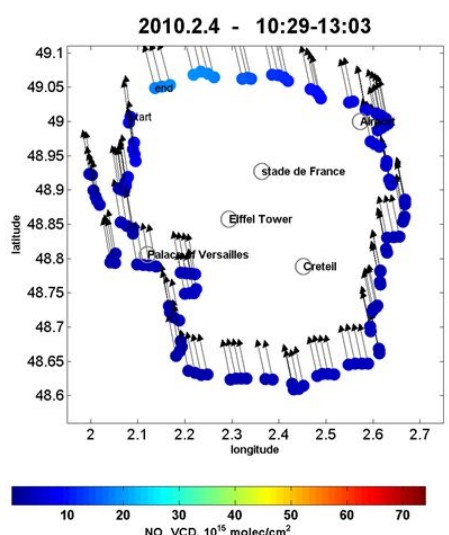

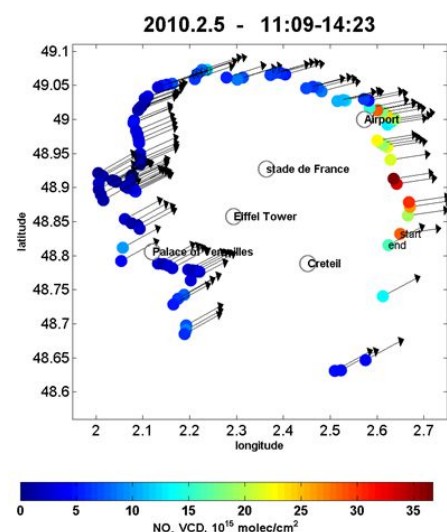

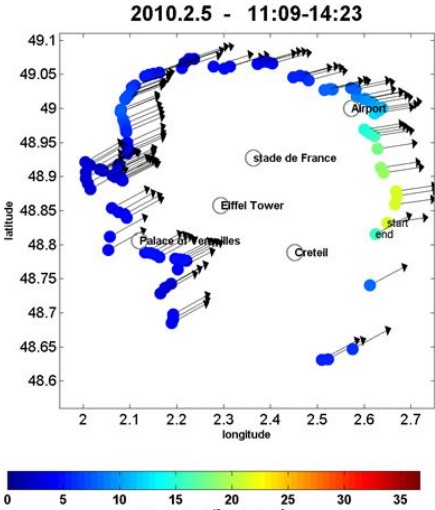





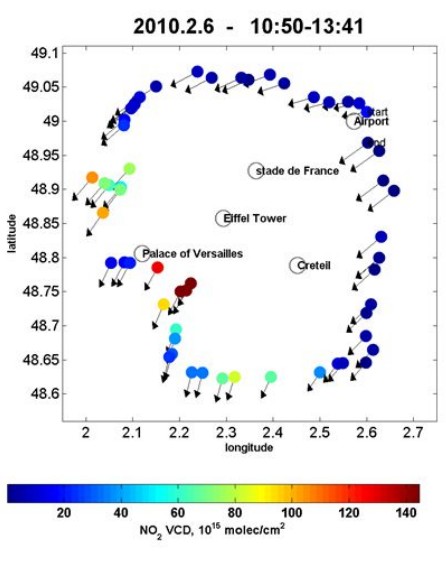
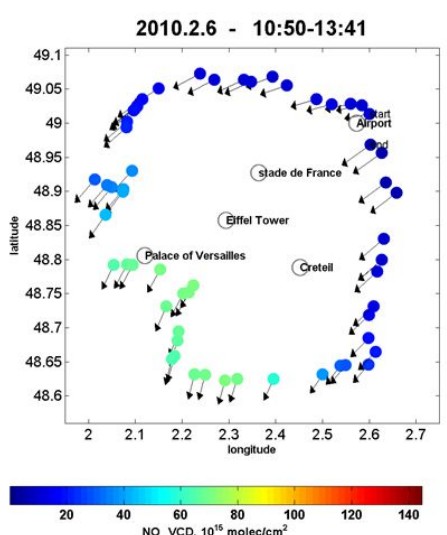

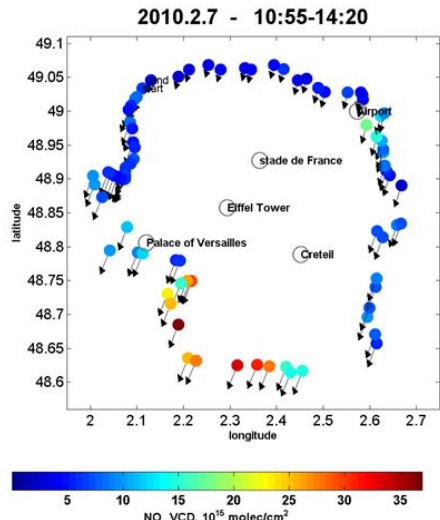
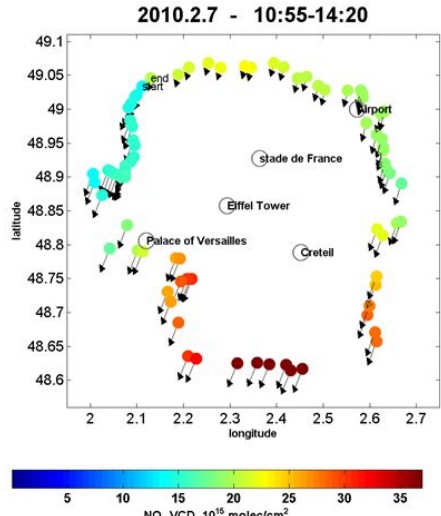



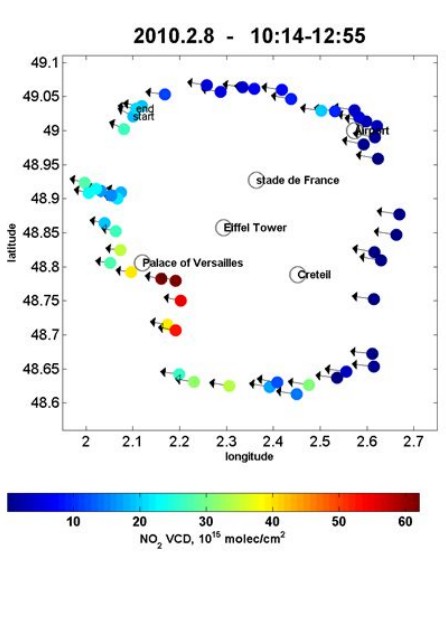

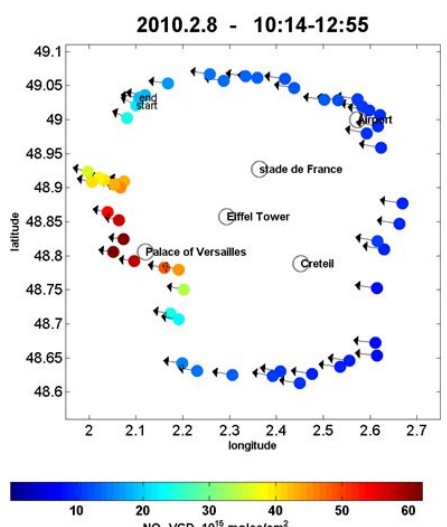

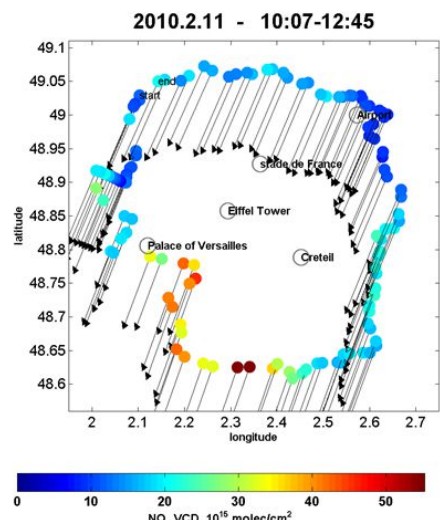

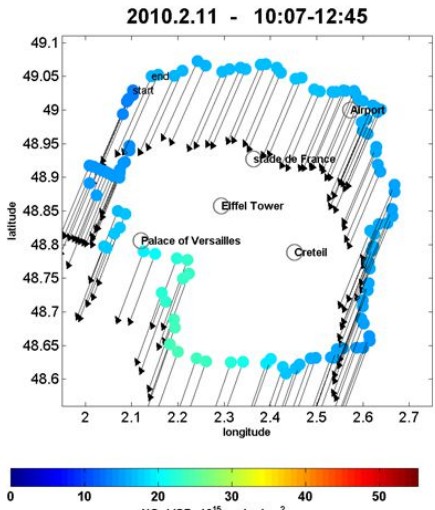





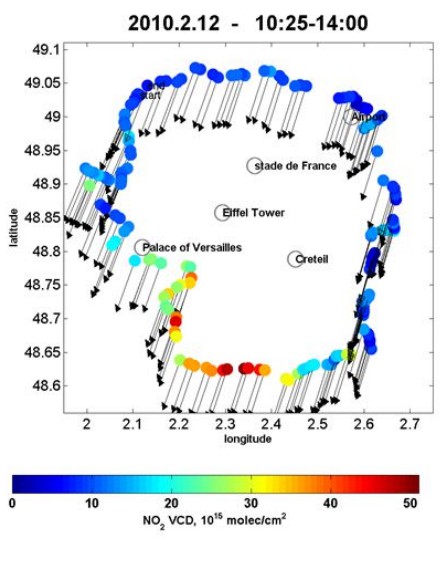

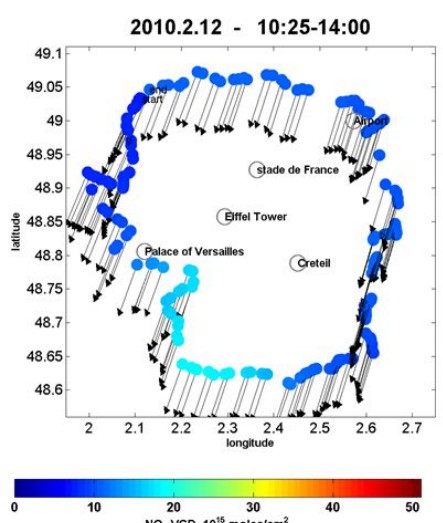

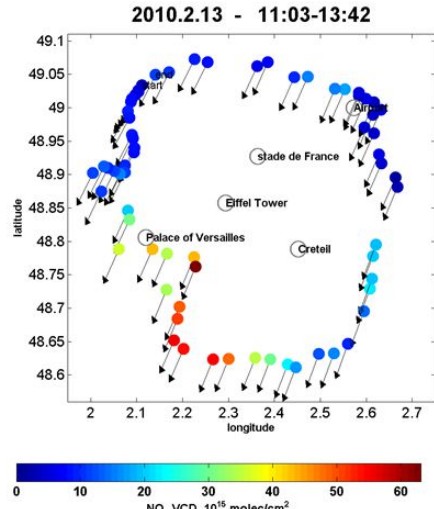

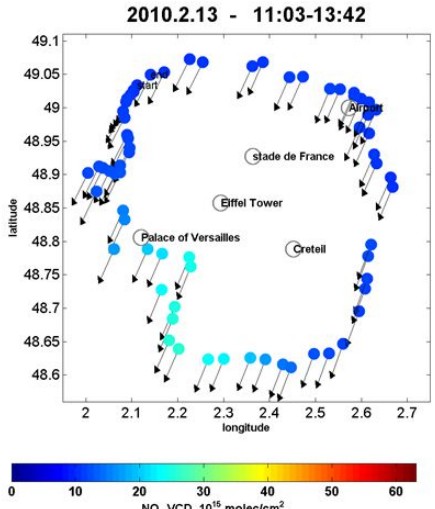




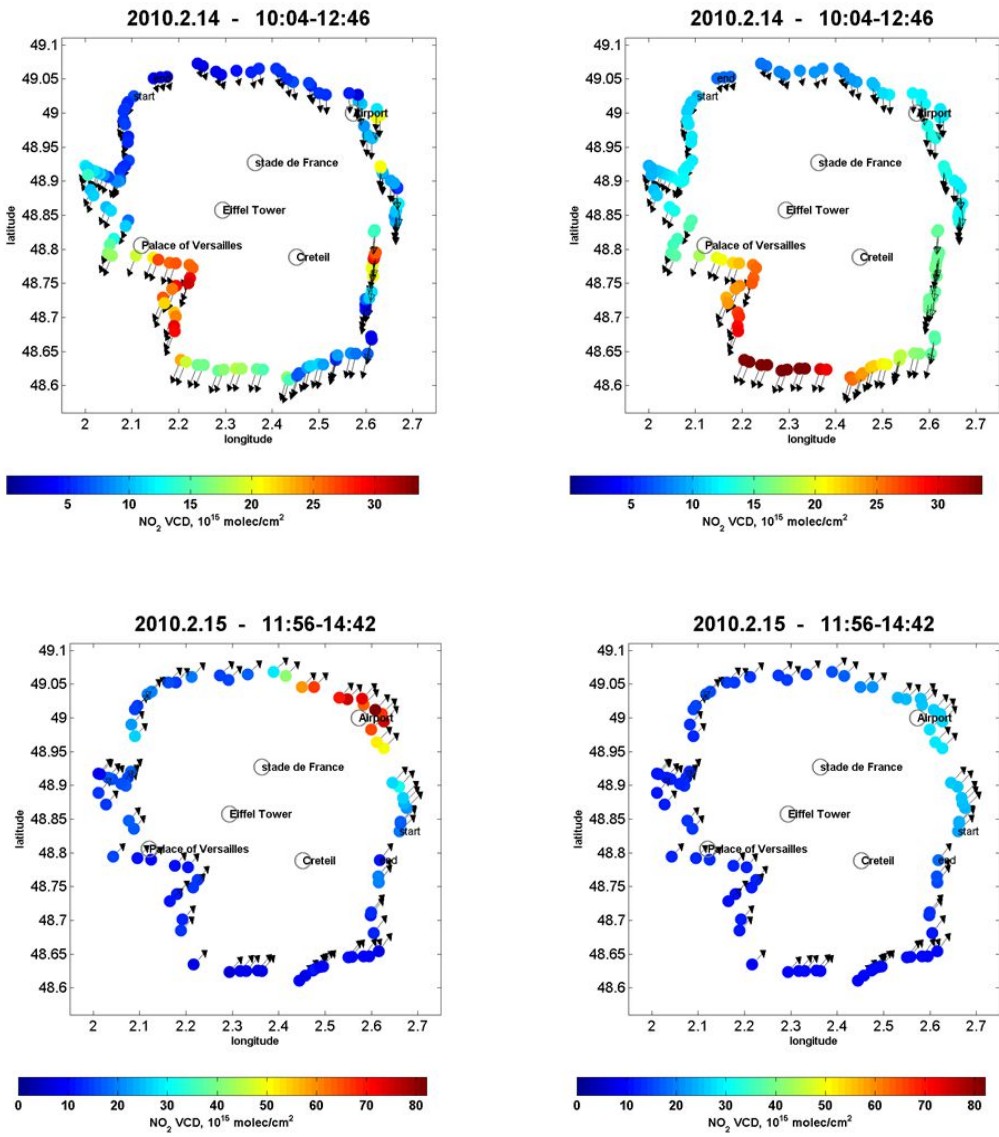

5    Fig. A9 NO$_2$ VCDs and wind vectors for all days of both campaigns. Left the results of the car-MAX-DOAS measurements, and right the corresponding model results are shown.