# Peer review of "Estimation of the Paris $NO_x$ Emissions from mobile MAX-DOAS observations and CHIMERE model simulations during the MEGAPOLI campaign using the closed integral method"

_Atmospheric Chemistry and Physics, 2016_

## Referee Comment (RC1) · Anonymous Referee #1 · 16 Dec 2016

General Comments I believe that this paper is highly acceptable and appropriate for publishing in ACP. This paper substantially contributes to scientific progress in the new field of mobile-MAX-DOAS, a useful new tool for atmospheric science and air quality monitoring. The large dataset (relative to any other previous publication) and the thorough and comprehensive data analyses presented in this paper support substantial conclusions about innovations to and examinations of mobile-MAX-DOAS techniques. These include addressing scientific questions such what are the optimal meteorological and measurement conditions for use of this method, the factors contributing the most to total error, potential modifications to the CIM method, and how consistent the

method was compared to modeling results. The authors contribute to the field by developing methods for error analysis and approaches to quality check data, essential for future utility of mobile-MAX-DOAS. The authors also successfully identify technique aspects requiring further exploration or improvement, such as the knowledge of the variability of the NO to NO2 partitioning ratio. This paper is of high scientific quality but may benefit from the authors adding some relatively minor clarifications to some of their methodology sections in order to optimize clarity and scientific reproducibility.

Scientific Comments

Section 3: In order to increase clarity (and reproducibility) when discussing methods of averaging or other manipulations of the hourly wind data obtained from the MM5 model, the authors could include more information such as: the maximum altitude range for the wind fields in the model, the size of the vertical altitude bins (e.g., what are the different heights referred to in section 4?) and the resolution of the model (i.e., is it also 3kmx3km?).

Section 4 Starting on line 5 the authors state that the wind speed and direction are averaged over the measurement area but it is not totally clear whether that refers to averaging in both the horizontal and the vertical or just the horizontal covering the measurement circle area. In general, it would help to improve clarity by specifying what is meant by "average wind" (temporally over a particular period and/or spatially over a specific vertical or horizontal distance) if the term means different things in different sections. It would be potentially helpful to specify more why the wind data are weighted by exponentially decreasing profiles and what actual form of equation was used (e.g., does the equation have any coefficients or variables other than scale height and altitude?). Also, it is unclear whether the MM5 vertical wind profiles are interpolated into a continuous profile from the MM5 altitude bins or averaged to a single value in the vertical before an exponential profile is applied. The authors state that these exponential vertical profiles account for different vertical mixing conditions but it is unclear if this could have been applied to the VCD instead since the exponential profile

appears to account for the fact that the NO2 is unlikely to be uniform in concentration with altitude in the boundary layer. This is related to wind but also to the NOx sources being predominantly (presumably) surface-based. On Line 19 it is potentially unclear as to how Fig. A1 shows that the stated assumptions are not necessarily valid.

Section 4.2 When discussing the comparison of clock-wise vs. counter-clockwise calculations of emission flux, it may be helpful for the reader to be reminded that when there is a gap the last VCD in the direction of calculation is used as the correct VCD for the gap segment (from previously published papers) and that this can contribute to the difference between the two calculations (unless this method was not used and then specify the new method).

Section 4.5 In section 4.5 the authors could chose to briefly address how homogenous the NOx sources are across Paris (e.g., major point source locations vs. high concentration road traffic/highways).

Section 4.6 In 4.6 on page 7, line 32: it may be useful within the error contribution discussion for the authors to address whether diurnal trends could introduce significant error when the time difference between measurements of influx and outflux is large. For example, if measurements start during a diurnal NOx emission peak but the emission rates decrease significantly well before the circle is completed (or vice versa). If this is the case, how would the authors determine what time period the resulting calculated emission value are representative of?

Section 4.6.1 The methodology in this section may benefit from greater clarity with some additional information. Starting on line 22: does this "error of F" refer to an estimate of the standard deviation of all the VCDs from an entire, single circle or for a specified segment of the measurement circle? This section may benefit from the authors also defining what is meant by a "single summand" and the difference between $\Delta$VCD and $\Delta$VCDi. $\Delta$VCDi is slightly confusing in the sense that is there not only one VCD derived for each measurement location i?

Technical Corrections

Title: The authors may choose to add "during the MEGAPOLI campaign" to the title so that it most clearly reflects the paper contents. Adding something about "examination of error and optimal conditions" may also help since these are important contributions to the field. I suggest writing emissions "from" rather than emissions "for" and writing measurements were performed "in" large circles rather than "at" or "on" for maximum clarity and grammatical correctness throughout. In general, when explaining data analyses completed or methodology used, use of the past tense is most correct (e.g., measurements "were" rather than "are" performed).

Abstract Line 15: add NO2 to influx into and outflux out of the encircled area for maximum clarity. Line 16: "The difference of both fluxes represents the total emission" could be changed to "The difference between the influx and outflux represent the total emission" for increased clarity. Line 22: It may be helpful to specify or give examples of "uncertainties" to minimize ambiguity. Line 25: There is an extra "p" in "developed" Section 3 Line 11/12: missing an "and" between by month and by source sector. Line 18: (typo) pick "a" or "the" in the sentence starting with "In Figure A1

Section 4 Page 4 line 4: "I" needs to be small-caps. There may be other instances of this in other parts of the paper. In this section the authors refer to A2 when referring to comparison of averaged wind speed and directions during periods of MAX-DOAS measurements yet the diagram shows NO2 profiles rather than wind data. I think this should be referring to figure A1.

Section 4.1 Page 4 line 10: missing a close bracket at the end of the sentence. Page 4 line 28/29: The NOx layer scale heights or layer heights?

Section 4.2 Page 5 line 5: Lowercase "I" needed.

Section 4.4 Page 6 line 14: comma needs to be replaced with a period. Page 6 sentence starting line 27: it is unclear what the "difference" is; this sentence may benefit

from rewording to improve clarify and (e.g., which VCDs are subtracted from which?)

Section 4.5 Page 7 line 19: typo, second last word in sentence.

Section 4.6.3 Line 19: typo "accounted" Line 28: specify the relative differences in "what" (e.g., NOx emissions)

Section 5 Line 33: typo, need the word 'beginning'

Figures and Tables Fig. 3 Add to the caption: during the time of measurement of the entire single circle.

Table1 – For large wind variability: Does "relative deviation of wind speed >30%" refer to a standard deviation using all the wind speeds during the measurement period or between the smallest and largest wind speeds for individual pairs?

---

## Referee Comment (RC2) · Anonymous Referee #2 · 18 Jan 2017

In this paper, Shaiganfar et al. Report on a series of car-based DOAS measurements of NO2 around Paris which they use to estimate NOx emissions of that city. The manuscript describes the measurements and approach to emission estimation and discusses the different contributions to the overall uncertainties of the derived NOx fluxes. It then applies the same flux estimation method to simulations of the CHIMERE model using the sampling of the measurements and compares the results to the integrated emission flux used in the model. Finally, emissions are derived from the measurements on 18 days and compared to the emissions from the TNO / AirParif inventory.

[Figure]

The paper is clearly structured, well written, reports on an interesting type of measurements and provides relevant emission estimates for Paris. The detailed error discussion provided is important for the application of similar measurements in other regions and will be useful for future measurements. My only major concern with this paper is that it mainly discusses the method used and its uncertainties and spends little time on the results and their implications. A journal such as AMT would therefore have been a better place for this manuscript. I therefore recommend publication of this paper only after strengthening the results and discussions part.

**Major point**

The paper reports measurements of NOx emission fluxes for Paris on 18 days and compares them to the TNO / AirParif emissions. The results as shown in Fig. 17 indicate good agreement between the two quantities on many days, but also large differences on other days. In particular in January / February, the car-DOAS based estimates show large day-to-day variations and much larger values than the emission inventory. This raises two questions:

1. Is it plausible that NOx emissions in Paris change by a factor of two between January 19 and February 11? The emission inventory suggests the same value for both days, and considering the fact that traffic is the dominant NOx source in Paris, what could be the origin of all the additional NOx? Or is this a problem of the measurements / method? The latter is not suggested by the results of the application of CIM to the model data, so this is a bit of a mystery.

2. Is it realistic that the TNO / AirParif NOx emission inventory is off by a factor of three as it appears from the last 4 days of measurements shown?

I think these two points deserve more discussion and analysis.

**Minor points**

- Section 3, line 9: The Figure in the Appendix referenced to does not exist

- Section 4.1, line 19: Wrong Figure number in Appendix

- Section 4.1, line 25: Wrong Figure number in Appendix

- Section 4.3, line 26: Is it expected that the emission flux depends on the largest values? And isn't that maybe a problem indicating that the car DOAS measurements are affected by close-by local sources more than they should?

- Section 4.3: Please mention and briefly discuss somewhere that you apply partitioning (and life time correction) to columns although strictly speaking this is something to be done on height levels.

- Section 4.6.1., line 19: simply => simple

- Section 4.6.3., line 19: Last sentence of paragraph is unclear, please reformulate

- Section 5: The definition of times for CIM application to the CHIMERE data is unclear to me – why did you not just use the times of the measurements? Using the time of the maximum measurement appears arbitrary to me but I may be missing the important point here. Please explain.

- Section 5 last paragraph: I find the discussion of weekend effects confusing – in Fig. 17. We can clearly see the weekend effect in the emissions but not in the CHIMERE.CIM values. Therefore, this is not a result of changes in domestic heating but just random uncertainties introduced by the method and sampling. Please re-consider.

- Section 5, last lines: What is the logic of only showing data with small differences between TNO and CIM values here? I could understand if only data without obvious problems were used, but the other values should appear in this figure in my opinion. Please re-consider.

- Section 6, last paragraph: It is noted twice that a similar ratio is found between CHIMERE VCs and observed VCs on the one hand and the emissions on the other hand. I think this is to be expected considering the way the emissions are determined from the columns which assumes a nearly linear relationship (excluding life time and partitioning corrections).

- Conclusions and perspectives: This section is mainly a summary and in parts identical to the abstract. As mentioned in the major comment, I think more focus should be on the results.

- Conclusions and perspectives: I do not agree with the statement, that the large number of measurements was used to test the applicability of CIM under various atmospheric conditions. Actually, all the tests were performed on the model data which could have been done without measurements by just assuming certain measurement routes and patterns. The data themselves are only used for emission estimates which is of course very interesting.

- Figure 2: I'm surprised that I cannot see the effect of daylight saving time in the diurnal emission pattern

- Figure 2: Are the emission values in the map given per 3 x 3 km2 pixel? In the caption, it is said that they are averaged over this area but I assume they are summed up?

---

## Author Comment (AC1) · 18 Apr 2017

Reply to Reviewer #1

Before we respond to the individual comments of the reviewer we give a short overview about the most important changes compared to the previous version fo our manuscrip:

A) The diurnal cycle of emissions (Fig. 2) was corrected: local time => UTC. Accordingly, the upscaling to the daily average emissions was corrected and Figures 14 – 17, Fig. A6, Table 4, and the text were updated. The new upscaling caused slight changes compared to the previous version: => consisteny of Chimere emission in/out

is enhanced => overall most daily averaged values decreased

B) A discussion about 'special gaps' was added to section 4.2. Such gaps are charac­terised by large differences between the start and end points of a circle. An example for such a measurement (from 4 February 2010) was added to Fig. 4 (right). The following text was added at the end of section 4.2: 'In Fig. 4 (right) an example for measurements without an obvious gap is shown. However, on that day a large differ­ence between the NO2 VCD between the start and end locations of the circle is found indicating that during the period of the measurements the NOx distribution around the location of the maximum outflux has changed significantly. Obviously, the NOx emis­sions derived from these measurements are subject to large uncertainties and are thus also skipped from the set of measurements considered for the comparison to the input emissions (section 6)'.

C) We added more discussion on the reasons for discrepancy between input emissios and car-MAX-DOAS results. Here two (related) aspects are important: -the rather high day to day variability of the car MAX-DOAS results -the enhanced seasonal cycle of the car MAX-DOAS results. We discuss both points in detail now in the conclusions. There the followoing text was added:

'Here it is interesting to note that a high day to day variability was also found by Petetin et al. (2015). For most of the measurement derived emission results, the day to day variability is within the range of the uncertainties, especially in summer. Thus we con­clude that this variability simply reflects the uncertainty range of the measurements. However, for several days at the end of the winter measurement campaign on mid-February, significantly enhanced values were found compared to the other winter days. These days are also the reason for the rather high average values derived from the car MAX-DOAS measurements in winter. If these days are excluded, a similar ratio (1.4) of the NOx emissions derived from car MAX-DOAS or CHIMERE as in summer (1.5) is found. Interestingly, for these days the temperature was low (-4°C to −1°C) indicating that the high emissions might be related to these low temperatures (see Fig. 18) The

following effects might be responsible for enhanced NOx emissions on cold days: a) Residential heating According to Fig. 2 domestic heating contributes about 25% to the total NOx emissions in winter. If one assumes a factor of two variability between cold and warm (less cold) winter days (see e.g. Terrenoire et al., 2015), it becomes clear that the variability of the NOx emissions from residential combustion alone can only explain a part of the increase of about a factor of two found for the cold days. b) Temperature dependence of catalytic converters During winter time, NOx emissions from traffic contribute about a half to the total NOx emissions. Under cold conditions, three way catalytic converters for gasoline cars work less well, and they take longer time to reach to an optimized way of working for diesel cars (the cold start effect). It is probable that this effect leads to increased NOx emissions on cold days, but this additional emission is difficult to quantify. c) It is known that in the past during cold periods an older 250MW coal-fired power plant was temporarily restarted to meet the additional demand for electrical heating in the city. Several other fuel or gas driven combustion turbines can also be activated during periods of increased energy demand. On an annual basis such temporarily operating facilities would not add much to the annual total emissions but during episodes it could be important. Instead of being spread out over the year, the emissions would have to be allocated to a much smaller number of operation days causing the emissions during selective periods to be much higher than annual averaged, and on other moments to be zero. Unfortunately, we have no access to operation days for such facilities and cannot confirm that this contributed also during the February episode discussed in this paper.'

General Comments I believe that this paper is highly acceptable and appropriate for publishing in ACP. This paper substantially contributes to scientific progress in the new field of mobile-MAX-DOAS, a useful new tool for atmospheric science and air quality monitoring. The large dataset (relative to any other previous publication) and the thorough and comprehensive data analyses presented in this paper support substantial conclusions about innovations to and examinations of mobile-MAX-DOAS techniques. These include addressing scientific questions such what are the optimal meteorological and measurement conditions for use of this method, the factors contributing the most to total error, potential modifications to the CIM method, and how consistent the method was compared to modeling results. The authors contribute to the field by developing methods for error analysis and approaches to quality check data, essential for future utility of mobile-MAX-DOAS. The authors also successfully identify technique aspects requiring further exploration or improvement, such as the knowledge of the variability of the NO to NO2 partitioning ratio. This paper is of high scientific quality but may benefit from the authors adding some relatively minor clarifications to some of their methodology sections in order to optimize clarity and scientific reproducibility.

Author reply: We thank the reviewer for the positive assessment.

Scientific Comments

Section 3: In order to increase clarity (and reproducibility) when discussing methods of averaging or other manipulations of the hourly wind data obtained from the MM5 model, the authors could include more information such as: the maximum altitude range for the wind fields in the model, the size of the vertical altitude bins (e.g., what are the different heights referred to in section 4?) and the resolution of the model (i.e., is it also 3kmx3km?).

Author reply: In section 3 (and also Fig. 2), we replaced 'averaged' by 'summed up'. In section, 3, already the altitude range (up to 5 km) and the horizontal resolution ($3x3km^2$) of the model simulations were stated.

In section 4 we added the information that the wind data between the surface and 1000m are used (weighted by an exponetial profile with either 300m or 500m).

In the revised version we also mention the exact way in which the wind data are averaged for the individual circles: '...we calculated the average wind speed and direction for all individual locations and times of the car MAX-DOAS measurements along the circle'.

[Figure]

Section 4 Starting on line 5 the authors state that the wind speed and direction are averaged over the measurement area but it is not totally clear whether that refers to averaging in both the horizontal and the vertical or just the horizontal covering the measurement circle area. Author reply: In the text in section 4 describing the averaging in time and location we added the following hint '(for the vertical averaging, see below)' to the end of section 4, where the vertical averaging is described.

In general, it would help to improve clarity by specifying what is meant by "average wind" (temporally over a particular period and/or spatially over a specific vertical or horizontal distance) if the term means different things in different sections.

Author reply: We believe that with the changes described above, in the revised version of the manuscript all necessary information about the averaging process of the wind data is provided (see above).

It would be potentially helpful to specify more why the wind data are weighted by exponentially decreasing profiles and what actual form of equation was used (e.g., does the equation have any coefficients or variables other than scale height and altitude?).

Author reply: To make the motivation for our procedure more clear, we added the following information to section 4: 'Both wind speed and direction vary systematically with altitude (see e.g. Fig. 3). Thus a choice has to be made, in which altitude range most NOx is probably situated, because the wind data for this altitude range determine the NOx flux. Since our measurements were performed close to the NOx emission sources, and since most NOx emission sources are located close to the surface, we assume expontentially decreasing NOx concentration profiles with scale heights of 300 m (winter) and 500 m (summer). We added an equation describing the vertical averaging (new equation 3) We also added the following information at the end of section 4.1: 'Here it should also be noted that the exact choice of the scale height is not critical: changes of the scale heights between 200 and 700m usually lead to differences of the wind speed <0.5m/s and wind direction $<5°$. The errors of the derived NOx emissions

associated with uncertainties of the wind speed and direction are quantified in section 4.6.3'.

Also, it is unclear whether the MM5 vertical wind profiles are interpolated into a continuous profile from the MM5 altitude bins or averaged to a single value in the vertical before an exponential profile is applied.

Author reply: The exact procedure is now described in the new eq. 3.

The authors state that these exponential vertical profiles account for different vertical mixing conditions but it is unclear if this could have been applied to the VCD instead since the exponential profile appears to account for the fact that the $NO_2$ is unlikely to be uniform in concentration with altitude in the boundary layer. This is related to wind but also to the $NO_x$ sources being predominantly (presumably) surface-based.

Author reply: We added more information to make the motivation for our approach more clear (see replies to comments above).

On Line 19 it is potentially unclear as to how Fig. A1 shows that the stated assumptions are not necessarily valid.

Author reply: Here reference to Fig. A2 should have been made. This is corrected now.

Section 4.2 When discussing the comparison of clock-wise vs. counter-clockwise calculations of emission flux, it may be helpful for the reader to be reminded that when there is a gap the last VCD in the direction of calculation is used as the correct VCD for the gap segment (from previously published papers) and that this can contribute to the difference between the two calculations (unless this method was not used and then specify the new method).

Author reply: We checked the text in section 4.2 and found that the provided description should be sufficiently clear: 'Since the values of the wind speed and direction in equation 2 are determined for the location of measurement i, but the distance $\Delta s_i$ is determined between measurement i and i+1, the direction for which the sum is calculated

leads to a difference in the derived total NO2 flux.'

Section 4.5 In section 4.5 the authors could chose to briefly address how homogenous the NOx sources are across Paris (e.g., major point source locations vs. high concentration road traffic/highways).

Author reply: We added the locations and types of the strongest point sources in the new table 1. We added the following text at the end of section 4.5: 'The location and type of the strongest point sources is presented in Table 1.'.

Section 4.6 In 4.6 on page 7, line 32: it may be useful within the error contribution discussion for the authors to address whether diurnal trends could introduce significant error when the time difference between measurements of influx and outflux is large. For example, if measurements start during a diurnal NOx emission peak but the emission rates decrease significantly well before the circle is completed (or vice versa). If this is the case, how would the authors determine what time period the resulting calculated emission value are representative of?

Author reply: At the beginning of section 4.6 (directly before section 4.6.1) it is already stated: 'It has, however, to be taken into account that the derived NOx emissions are only representative for a specific time period of the day (mainly depending on wind speed and the diameter of the driving circle, see also section 5)'. In section 5, we added the following information: 'Here it is interesting to note that for the cases considerd here time variations of $\pm 1$ h lead to changes of the respective input emissions of 2% to 15%.'.

Section 4.6.1 The methodology in this section may benefit from greater clarity with some additional information. Starting on line 22: does this "error of F" refer to an estimate of the standard deviation of all the VCDs from an entire, single circle or for a specified segment of the measurement circle?

Author reply: We checked the text in section 4.6.1 and found that the information should

be sufficient and correct. In order to avoid confusion, we slightly modified the text: 'In order to estimate the error of F due to gaps, we use the following approach: First we estimate the uncertainty of VCD simply by the standard deviation of all measurements VCDi:'.

This section may benefit from the authors also defining what is meant by a "single summand" and the difference between _VCD and _VCDi. _VCDi is slightly confusing in the sense that is there not only one VCD derived for each measurement location i?

Author reply: We checked the text, and found that the information in section 4.6.1 is consistent with the definition of the respective quantities in equation 2.

Technical Corrections

Title: The authors may choose to add "during the MEGAPOLI campaign" to the title so that it most clearly reflects the paper contents. Adding something about "examination of error and optimal conditions" may also help since these are important contributions to the field.

Author reply: We added "during the MEGAPOLI campaign" to the title. However, we did not add more text to the title, because it is already quite long.

I suggest writing emissions "from" rather than emissions "for" and writing measurements were performed "in" large circles rather than "at" or "on" for maximum clarity and grammatical correctness throughout.

Author reply: corrected

In general, when explaining data analyses completed or methodology used, use of the past tense is most correct (e.g., measurements "were" rather than "are" performed).

Author reply: We changed to 'past tense' for such parts throughout the manuscript.

Abstract Line 15: add NO2 to influx into and outflux out of the encircled area for maximum clarity.

Author reply: The suggested text was added.

Line 16: "The difference of both fluxes represents the total emission" could be changed to "The difference between the influx and outflux represent the total emission" for increased clarity.

Author reply: The text was changed.

Line 22: It may be helpful to specify or give examples of "uncertainties" to minimize ambiguity.

Author reply: We added the information that '…., which typically ranges between 30% and 50%'

Line 25: There is an extra "p" in "developed"

Author reply: corrected

Section 3 Line 11/12: missing an "and" between by month and by source sector.

Author reply: 'and' was added

Line 18: (typo) pick "a" or "the" in the sentence starting with "In Figure A1

Author reply: corrected

Section 4 Page 4 line 4: "I" needs to be small-caps. There may be other instances of this in other parts of the paper.

Author reply: corrected

In this section the authors refer to A2 when referring to comparison of averaged wind speed and directions during periods of MAX-DOAS measurements yet the diagram shows NO2 profiles rather than wind data. I think this should be referring to figure A1.

Author reply: Yes, it should be Fig. A1. The text was corrected

Section 4.1 Page 4 line 10: missing a close bracket at the end of the sentence.

Author reply: A brackett was added.

Page 4 line 28/29: The NOx layer scale heights or layer heights?

Author reply: The text was changed to 'when weighted with the exponential NOx profiles with scale heights of 300 m and 500 m, respectively.'

Section 4.2 Page 5 line 5: Lowercase "I" needed.

Author reply: corrected

Section 4.4 Page 6 line 14: comma needs to be replaced with a period.

Author reply: corrected

Page 6 sentence starting line 27: it is unclear what the "difference" is; this sentence may benefit from rewording to improve clarify and (e.g., which VCDs are subtracted from which?)

Author reply: The text is changed to '...the lifetime correction was only applied to the enhancement of the NO2 VCDs over the minimum NO2 VCDs at the upwind side.'.

Section 4.5 Page 7 line 19: typo, second last word in sentence.

Author reply: corrected

Section 4.6.3 Line 19: typo "accounted"

Author reply: corrected

Line 28: specify the relative differences in "what" (e.g., NOx emissions)

Author reply: The text is changed to 'Fig. 12 displays the relative differences of the derived NOx emissions with either averaged or spatio-temporally varying wind fields for all days...'.

Section 5 Line 33: typo, need the word 'beginning'

[Figure]

Author reply: corrected

Figures and Tables Fig. 3 Add to the caption: during the time of measurement of the entire single circle.

Author reply: The information was added.

Table1 – For large wind variability: Does "relative deviation of wind speed >30%" refer to a standard deviation using all the wind speeds during the measurement period or between the smallest and largest wind speeds for individual pairs?

Author reply: We added '(vmax – vmin)' to the table.

―――――――――――――――――――

---

## Author Comment (AC2) · 18 Apr 2017

Reply to reviewer #2

Before we respond to the individual comments of the reviewer we give a short overview about the most important changes compared to the previous version fo our manuscrip:

A) The diurnal cycle of emissions (Fig. 2) was corrected: local time => UTC. Accordingly, the upscaling to the daily average emissions was corrected and Figures 14 – 17, Fig. A6, Table 4, and the text were updated. The new upscaling caused slight changes compared to the previous version: => consisteny of Chimere emission in/out

is enhanced => overall most daily averaged values decreased

B) A discussion about 'special gaps' was added to section 4.2. Such gaps are characterised by large differences between the start and end points of a circle. An example for such a measurement (from 4 February 2010) was added to Fig. 4 (right). The following text was added at the end of section 4.2: 'In Fig. 4 (right) an example for measurements without an obvious gap is shown. However, on that day a large difference between the NO2 VCD between the start and end locations of the circle is found indicating that during the period of the measurements the NOx distribution around the location of the maximum outflux has changed significantly. Obviously, the NOx emissions derived from these measurements are subject to large uncertainties and are thus also skipped from the set of measurements considered for the comparison to the input emissions (section 6)'.

C) We added more discussion on the reasons for discrepancy between input emissios and car-MAX-DOAS results. Here two (related) aspects are important: -the rather high day to day variability of the car MAX-DOAS results -the enhanced seasonal cycle of the car MAX-DOAS results. We discuss both points in detail now in the conclusions. There the followoing text was added:

'Here it is interesting to note that a high day to day variability was also found by Petetin et al. (2015). For most of the measurement derived emission results, the day to day variability is within the range of the uncertainties, especially in summer. Thus we conclude that this variability simply reflects the uncertainty range of the measurements. However, for several days at the end of the winter measurement campaign on mid-February, significantly enhanced values were found compared to the other winter days. These days are also the reason for the rather high average values derived from the car MAX-DOAS measurements in winter. If these days are excluded, a similar ratio (1.4) of the NOx emissions derived from car MAX-DOAS or CHIMERE as in summer (1.5) is found. Interestingly, for these days the temperature was low (-4°C to –1°C) indicating that the high emissions might be related to these low temperatures (see Fig. 18) The

following effects might be responsible for enhanced NOx emissions on cold days: a) Residential heating According to Fig. 2 domestic heating contributes about 25% to the total NOx emissions in winter. If one assumes a factor of two variability between cold and warm (less cold) winter days (see e.g. Terrenoire et al., 2015), it becomes clear that the variability of the NOx emissions from residential combustion alone can only explain a part of the increase of about a factor of two found for the cold days. b) Temperature dependence of catalytic converters During winter time, NOx emissions from traffic contribute about a half to the total NOx emissions. Under cold conditions, three way catalytic converters for gasoline cars work less well, and they take longer time to reach to an optimized way of working for diesel cars (the cold start effect). It is probable that this effect leads to increased NOx emissions on cold days, but this additional emission is difficult to quantify. c) It is known that in the past during cold periods an older 250MW coal-fired power plant was temporarily restarted to meet the additional demand for electrical heating in the city. Several other fuel or gas driven combustion turbines can also be activated during periods of increased energy demand. On an annual basis such temporarily operating facilities would not add much to the annual total emissions but during episodes it could be important. Instead of being spread out over the year, the emissions would have to be allocated to a much smaller number of operation days causing the emissions during selective periods to be much higher than annual averaged, and on other moments to be zero. Unfortunately, we have no access to operation days for such facilities and cannot confirm that this contributed also during the February episode discussed in this paper.'

In this paper, Shaiganfar et al. Report on a series of car-based DOAS measurements of NO2 around Paris which they use to estimate NOx emissions of that city. The manuscript describes the measurements and approach to emission estimation and discusses the different contributions to the overall uncertainties of the derived NOx fluxes. It then applies the same flux estimation method to simulations of the CHIMERE model using the sampling of the measurements and compares the results to the integrated emission flux used in the model. Finally, emissions are derived from the measurements

on 18 days and compared to the emissions from the TNO / AirParif inventory. The paper is clearly structured, well written, reports on an interesting type of measurements and provides relevant emission estimates for Paris. The detailed error discussion provided is important for the application of similar measurements in other regions and will be useful for future measurements. My only major concern with this paper is that it mainly discusses the method used and its uncertainties and spends little time on the results and their implications. A journal such as AMT would therefore have been a better place for this manuscript. I therefore recommend publication of this paper only after strengthening the results and discussions part.

Author reply: We thank the reviewer for the positive assessment. We are aware of the fact that a large part of the paper describes technical aspects of the car-MAX-DOAS measurements. We therefore also asked ourselves whether submission to AMT would be more appropriate. However, in addition to the technical aspects, the paper provides the first detailed comparison of the experimentally derived NOx emissions from Paris to existing emission inventories and model simulations (during extended measurement campaigns). We regard the comparison results as important information for a wider community than only the measurement experts. Thus in our opinion publication in ACP is well justified. In the revised version we spend more emphasis on the comparison results and discuss in more detail possible reasons for the discrepancies between the experimental results and the existing emission inventaries (see point C above).

Major point

The paper reports measurements of NOx emission fluxes for Paris on 18 days and compares them to the TNO / AirParif emissions. The results as shown in Fig. 17 indicate good agreement between the two quantities on many days, but also large differences on other days. In particular in January / February, the car-DOAS based estimates show large day-to-day variations and much larger values than the emission inventory. This raises two questions:

1. Is it plausible that NOx emissions in Paris change by a factor of two between January 19 and February 11? The emission inventory suggests the same value for both days, and considering the fact that traffic is the dominant NOx source in Paris, what could be the origin of all the additional NOx? Or is this a problem of the measurements / method? The latter is not suggested by the results of the application of CIM to the model data, so this is a bit of a mystery.

Author reply: We investigated possible reasons for the high values at the end of February and added a detailed discussion to the conclusions section (see general point C above).

2. Is it realistic that the TNO / AirParif NOx emission inventory is off by a factor of three as it appears from the last 4 days of measurements shown? I think these two points deserve more discussion and analysis.

Author reply: As stated above, these high values occur only during the cold period in mid February. For other days, the differences are much smaller (10% to 50%). Thus our conclusion is that changes in emission sources due to the cold temperatures are the most probable reason for the discrepancies (see general point C above)

Minor points

ć Section 3, line 9: The Figure in the Appendix referenced to does not exist

Author reply: Many thanks for this hint. We removed this sentence from the text.

ć Section 4.1, line 19: Wrong Figure number in Appendix

Author reply: The number was changed to ‘A2’

ć Section 4.1, line 25: Wrong Figure number in Appendix

Author reply: The number was changed to ‘A1’.

ć Section 4.3, line 26: Is it expected that the emission flux depends on the largest

values? And isn't that maybe a problem indicating that the car DOAS measurements are affected by close-by local sources more than they should?

Author reply: It is true that close to emission sources high concentrations (and also VCDs occur. But this is not a problem, because the potemtially high concentrations (VCDs) are exactly balanced by their small spatial extent.

ć Section 4.3: Please mention and briefly discuss somewhere that you apply partitioning (and life time correction) to columns although strictly speaking this is something to be done on height levels.

Author reply: In section 4.3 (partitioning correction) we added the information that the partitioning ratio from the model was calculated from the respective VCDs. Thus it actually is representative for the VCD. In section 4.4 (lifetime correction), we changed the text to: 'Here it should be noted that these lifetimes are rough assumptions, and on individual days large deviations from the assumed values might occur. Moreover, in a strict sense separate lifetime corrections should be applied for individual height layers. But especially for wind speeds above about 2 m/s, the effect of the limited lifetime of NOx and thus of the uncertainties of the assumed lifetimes are small (the correction factor is close to unity)'.

ć Section 4.6.1., line 19: simply => simple

Author reply: corrected

ć Section 4.6.3., line 19: Last sentence of paragraph is unclear, please reformulate

Author reply: corrected (account => accounted)

ć Section 5: The definition of times for CIM application to the CHIMERE data is unclear to me – why did you not just use the times of the measurements? Using the time of the maximum measurement appears arbitrary to me but I may be missing the important point here. Please explain.

Author reply: We added the following information to the text: 'By selecting this time period we take into account the average travel time of the pollued air masses until the location of the measurement.'

ć Section 5 last paragraph: I find the discussion of weekend effects confusing – in Fig. 17. We can clearly see the weekend effect In the emissions but not in the CHIMERE.CIM values. Therefore, this is not a result of changes in domestic heating but just random uncertainties introduced by the method and sampling. Please re-consider.

Author reply: We added the following information to the text: '. . ..not the lowest emissions are found, indicating that the variation of the NOx emissions derived from car MAX-DOAS is not dominated by the weekend effect.' We deleted the statement about the domestic heating in this section. The possible influence of domestic heating is discussed in more detail in section 6.

ć Section 5, last lines: What is the logic of only showing data with small differences between TNO and CIM values here? I could understand if only data without obvious problems were used, but the other values should appear in this figure in my opinion. Please re-consider.

Author reply: We agree that in this figure this selection makes no sense. Therefore we included all data in the updated figure 15. inter-friendly versioniscussion paper ć Section 6, last paragraph: It is noted twice that a similar ratio is found between CHIMERE VCs and observed VCs on the one hand and the emissions on the other hand. I think this is to be expected considering the way the emissions are determined from the columns which assumes a nearly linear relationship (excluding life time and partitioning corrections).

Author reply: We deleted this staement at the end of section 6. However, we prefer to keep this statement in the conclusions, because this consistency between the comparison of the CIM results and the direct comparison of the NO2 VCDs serves as a

consistency check. In the conclusions we added the following information to the text: '...indicating that the differences between the measurements and the model simulations are not caused by the application of the CIM.'

ć Conclusions and perspectives: This section is mainly a summary and in parts identical to the abstract. As mentioned in the major comment, I think more focus should be on the results.

Author reply: We discuss the discrepancies between the experimental results and existing emission inventories in more detail. (see reply to general point above)

ć Conclusions and perspectives: I do not agree with the statement, that the large number of measurements was used to test the applicability of CIM under various atmospheric conditions. Actually, all the tests were performed on the model data which could have been done without measurements by just assuming certain measurement routes and patterns. The data themselves are only used for emission estimates which is of course very interesting.

Author reply: We agree that the important aspect here is that a large number of both car DOAS measurements and model simulations were available during the megapoli campaign. Thus we added '(together with the model results)' after 'the large number of measurements'.

ć Figure 2: I'm surprised that I cannot see the effect of daylight saving time in the diurnal emission pattern

Author reply: We thank the reviewer for this hint, which pointed our attention to a mistake we made (we mixed local time and universal time). The figure is corrected in the updated version of the manuscript (the time shift between summer and winter is now clearly visible).

ć Figure 2: Are the emission values in the map given per 3 x 3 km2 pixel? In the caption, it is said that they are averaged over this area but I assume they are summed

up?

Author reply: The text was changed to ‘summed up’.